# SPF-Portrait: Towards Pure Text-to-Portrait Customization with Semantic Pollution-Free Fine-Tuning

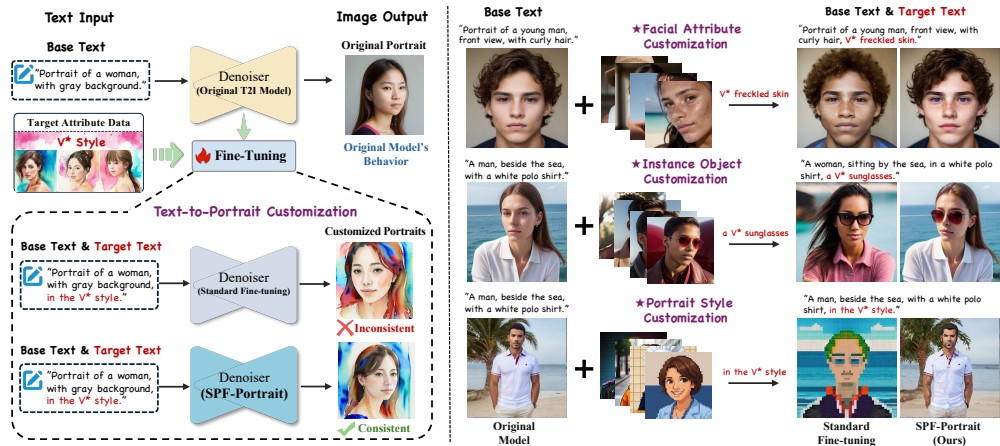

Figure 1: **Left:** The mainstream paradigm of text-to-portrait customization. **Right:** SPF-Portrait excellently achieves customized target semantics across various portrait attribute dimensions while significantly mitigating the disruption of the original model's behavior.

## Abstract

Fine-tuning a pre-trained text-to-image (T2I) diffusion model on a tailored portrait dataset is the mainstream method for text-to-portrait customization. However, existing methods often severely impact the original model's behavior (e.g., changes in ID, posture, layout, etc.) while customizing portrait. To address this issue, we propose **SPF-Portrait**, a pioneering work to achieve pure text-to-portrait customization, which necessitates direct text-conditioned personalized portrait generation and introduces differences purely through target attributes while preserving the original model's behavior before and after portrait customization. To eliminate the interference of conventional customization on the original model, SPF-Portrait designs an additional dual-path alignment stage after the standard fine-tuning. This stage introduces the pre-trained T2I diffusion model as a reference for the fine-tuned model and achieves behavioral alignment by contrastively constraining intermediate features in diffusion models between the dual paths. To accurately align target-unrelated attributes with the original behavior without affecting the effectiveness of the target response, we propose a novel Semantic-Aware Fine Control Map, which perceives the desired response region of target semantics to adaptively guide the alignment process, preventing over-alignment of the customized portrait with the original portrait. Furthermore, to improve the fidelity of target attribute, we introduce a novel target response enhancement mechanism that utilizes our proposed representation bias as a supervisory signal to mitigate the cross-modal discrepancy in direct text-image supervision, thereby reinforcing the performance of target attributes and the overall quality of the portraits. Extensive experiments demonstrate the superiority of our method.

## 1 Introduction

Text-to-portrait customization (Huang et al., 2023; Han et al., 2024; He et al., 2024) is widely researched due to its significant application in fields such as advertising and social media. As shown

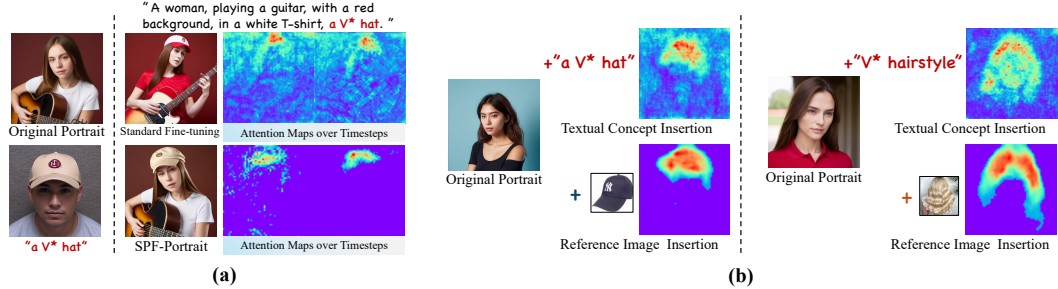

Figure 2: **(a) Visualization of the Attention Map** reflects the response regions of the target semantics 'a V* hat'. **(b) Comparison of Interference with Original Portrait.** In customization tasks based on T2I foundational models, the reference image insertion provides a more well-defined and decoupled condition to the basic text condition than the textual concept insertion. The fine-tuned model can better determine the target response region, limiting interference with other regions. To intuitively observe the difference, we simultaneously increase the visualization threshold of the attention map for the target condition.

in Fig. 1 (Left), fine-tuning on user-specific text-portrait image datasets (Rombach et al., 2022; Ramesh et al., 2021; Saharia et al., 2022; Esser et al.) is the mainstream paradigm for text-to-portrait customization. It allows pre-trained text-to-image (T2I) diffusion models to generate personalized portrait attributes by incorporating target semantics into base text conditions. However, existing fine-tuning methods primarily focus on improving the fidelity of target attributes, while neglecting the significant disruption to the original model's behavior during attribute customization (Guo et al., 2024), resulting in changes in target-unrelated attributes, such as the original portrait's identity, posture, background, layout, etc., as illustrated in Fig. 1 and Fig. 6. By visualizing the attention map in Fig. 2 (a), we reveal that this is because the fine-tuned diffusion model struggles to identify the response region of the target semantics ('a V* hat'), leading to its expansion into other areas unrelated to the target attributes. Consequently, when the fine-tuned model achieves the target semantics, it causes unexpected responses in other areas, disrupting the original behavior. We define this phenomenon as *Semantic Pollution*, which is detrimental and is often ignored before.

To address this issue, in this paper, we focus on a new task: **Pure Text-to-Portrait Customization** (a T2I Customization), which aims to provide direct text-conditioned portrait personalization while minimizing disruption to the original model caused by semantic pollution. Although previous work (Wang et al., 2024b; Ye et al., 2025) like PuLID (Guo et al., 2024) attempts to mitigate the impact on the original portrait caused by ID-image insertion (an I2I Customization). However, as analyzed in Fig. 2 (b), the destruction to the original portrait caused by textual concept insertion is more severe than image insertion, yet it has not been emphasized or specifically studied. Moreover, unlike those methods limited to ID customization, our approach enables convenient text-controlled portrait customization across fine-grained to global attributes, such as facial attributes, instance objects, and styles, shown in Fig. 1 (Right). Based on our task setting, related research utilizes LoRA (Hu et al., 2021) and its variants (Ding et al., 2023; Zhang et al., 2023b; Borse et al., 2024) to learn target semantics in additional low-rank subspaces, reducing the impact on original model. However, they rely solely on diffusion loss for learning without additional designed behavioral constraints, resulting in limited effectiveness. Another line of work attempts to purify the understanding of text and separate attributes from each other through embedding-level decoupling (Zhuang et al., 2024; Chen et al., 2024)) or regularization on attention maps (Wang et al., 2024c; Han et al., 2025). When applied to our task setting, they mitigate the influence on the original model for instance-level semantics customization, but fail to purely customize refined portrait attributes, like skin texture or hairstyle, as these fine-grained semantic concepts lack distinctive boundaries in the embedding space.

Given the limitations of existing approaches, we propose **SPF-Portrait** to achieve two specific objectives for pure text-to-portrait customization: *1)* effective and high-quality adaptation of T2I diffusion models to various customized portrait attributes and *2)* minimization of the disruption from semantic pollution to the original model. To initially obtain the T2I capability of the target attributes, we first fine-tune the pre-trained T2I diffusion model based on standard paradigm (Rombach et al., 2022). To eliminate semantic pollution, SPF-Portrait incorporates an additional stage, namely a dual-path alignment, which introduces the frozen pre-trained T2I diffusion model as the anchor of original behavior for the fine-tuned model. We contrastively constrain variant attention features and diffusion backbone features from the corresponding cross-attention layers between the dual paths. This enables the customized portrait to align both locally and globally with the original

model's behavior. Furthermore, to ensure that the alignment process only affects target-unrelated attributes, thereby avoiding the suppression of target attributes, we propose a novel Semantic-Aware Fine Control Map (SFCM), which accurately perceives the response regions of target semantics to spatially guide the alignment of intermediate features. Moreover, to enhance the fidelity of the target attributes, we propose an innovative response enhancement mechanism for the target semantics. By supervising the representation bias of target semantics between the one-step prediction and the ground truth image, we mitigate the cross-modal gaps inherent in direct text-image supervision, thereby enhancing the performance of target attributes while improving the overall quality of the image. Extensive experiments show that SPF-Portrait achieves state-of-the-art (SOTA) performance in preventing semantic pollution for pure text-to-portrait customization.

In summary, our contributions are as follows: • We propose a new task named pure text-to-portrait customization, which requires direct text-driven portrait personalization and purely inserts target portrait attributes while minimizing disruption to the original model's behavior. To achieve this, we develop SPF-Portrait, a pioneering work that addresses semantic pollution for pure customization with an additional dual-path alignment stage. • We introduce a novel Semantic-Aware Fine Control Map to spatially guide the alignment process, precisely aligning target-unrelated attributes with original portrait while ensuring the effective response of target semantics. • We design an innovative response enhancement mechanism to improve the fidelity of target attributes and overall quality of the image by alleviating cross-modal gaps in direct text-image supervision. • Extensive quantitative and qualitative experimental results demonstrate the superiority of our SPF-Portrait.

## 2 RELATED WORK

**Fine-tuning for Pure T2I Customization.** Numerous works (Zhang et al., 2023a; Wang et al., 2024b; Ruiz et al., 2023; Huang et al., 2024) extend existing T2I diffusion models (Rombach et al., 2022; Lin et al., 2024) to various customization tasks, primarily through fine-tuning. To mitigate the impact on the original model, PEFT-based methods (Wu et al., 2024; Zhang et al., 2023b; Borse et al., 2024) adapt to new concepts by introducing additional parameters outside of the original model. LoRA (Hu et al., 2021) achieves this through low-rank linear layers, while AdaLoRA (Zhang et al., 2023b) further enhances this by adaptively allocating low-rank budgets across layers based on learned importance scores. Related studies (Han et al., 2023; Qiu et al., 2023; Liu et al., 2023) also attempt to improve the preservation of prior knowledge during fine-tuning, for example, by focusing solely on fine-tuning key parameters through singular value decomposition and maintaining the orthogonality of weight matrices. Although these methods reduce disruption to the original model by preserving pre-trained knowledge, their reliance solely on diffusion loss lacks specific alignment constraints with original behavior, limiting their effectiveness of pure T2I customization.

**Decoupling Control for T2I Customization.** Decoupling control mechanisms aim to prevent the hindrance to textual control (Qi et al., 2024; Gao et al., 2024; Chang et al., 2024). To achieve decoupling within textual concepts, Magnet (Zhuang et al., 2024) and TEBopt (Chen et al., 2024) analyze and optimize condition embedding without requiring additional training. TokenCompose (Wang et al., 2024c) and STORM (Han et al., 2025) leverage attention map repositioning for precise spatial alignment. Although these decoupling methods are not initially designed for T2I customization tasks, their strategies are inherently beneficial for pure attribute customization and can be applied to our task. They can reduce instance-level attribute coupling in portrait customization but struggle with refined portrait attributes due to their lack of distinctiveness in the embedding space.

**Task Distinction with Text-driven Image Editing.** For image editing with a T2I model, (Brooks et al., 2023; Jiao et al., 2025; Kim et al., 2022) enable attribute manipulation through textual instructions, while (Ju et al., 2024; Kim et al., 2025) require mask input for precise, localized edits. Although integrating text-driven editing methods (Deutch et al., 2024; Wang et al., 2024a) into the T2I model pipeline can yield results comparable to ours. However, unlike these editing methods, which operate on a fixed image to adjust or refine existing content, we customize the T2I model itself to generate new target attributes from text while preserving its original generation behavior.

## 3 METHODOLOGY

SPF-Portrait can build upon both UNet-based and Diffusion Transformer-based (DiT-based) diffusion models for pure text-to-portrait customization. In this section, we detail our methodology based on the Unet architecture and provide the DiT-based implementation in the Appendix D.

Figure 3: **The Dual-Path Alignment Pipeline of SPF-Portrait.** The text in **blue** is the **Base text**, while those in **red** is the **Target text**. The upper path is the **Reference Path**, while the lower path is the **Response Path**.

## 3.1 PRELIMINARY

**T2I Diffusion Models** generate images based on text input through a forward diffusion process and a reverse denoising process (Ho et al., 2020; Saharia et al., 2022). The diffusion process follows the Markov chain to transform an image sample $x_0$ into noisy samples $x_{1:T}$ by adding Gaussian noise $\epsilon$ over $T$ steps. The denoising process employs a denoising model $\epsilon_\theta$ to predict the added noise using $x_t$, $t$, and textual conditions $y$ as inputs, where $\theta$ denotes the learnable parameters and $t \in [0, T]$ is the timestep of diffusion process. The optimization process can be described as:

$$\mathcal{L}_{diff} = \mathbb{E}_{x_0, \epsilon \sim \mathcal{N}(0,1), t}(\|\epsilon - \epsilon_\theta(x_t, t, E)\|_2^2), \tag{1}$$

where $E = \tau_{text}(y)$ denotes textual features, obtained from the textual conditions $y$ encoded by the text encoder $\tau_{text}$.

## 3.2 DUAL-PATH ALIGNMENT PIPELINE

SPF-Portrait accomplishes the two aforementioned objectives for pure text-to-portrait customization by two stages of training. **In the first stage**, we fine-tune the pre-trained T2I model on the user-specific text-portrait image dataset, following the standard paradigm (Rombach et al., 2022). It aims to achieve the preliminary adaptation of the model to target portrait attributes without considering interference with the original model. **In the second stage**, we design a dual-path pipeline and utilize contrastive alignment to eliminate semantic pollution from the first stage. Specifically, as shown in Fig. 3, the proposed dual paths including: (i) **Reference Path** is the frozen model loading from the original pre-trained T2I model. In contrastive alignment, it only takes base text $y_{base}$ (*"A woman in a brown jacket, with a curtain background, ......"*) as input, serving as a stable reference on behalf of the original model's behavior; and (ii) **Response Path** is the fine-tuned model initially resumed from the first stage. During the contrastive alignment, it is trainable and takes complete text: $[y_{base}, y_{tar}]$ as input. $[y_{base}, y_{tar}]$ represents the concatenated text prompt (*"A woman in a brown jacket, with a curtain background, ...... V\* hairstyle"*). The textual feature inputs for the two paths can be represented as:

$$\begin{cases} \text{Reference Path: } E_{base}^{ref} = \tau_{text}(y_{base}), \\ \\ \text{Response Path: } \begin{cases} E_{base}^{res} = \tau_{text}([y_{base}, y_{tar}])|_{y_{base}}, \\ E_{tar} = \tau_{text}([y_{base}, y_{tar}])|_{y_{tar}}, \end{cases} \end{cases} \tag{2}$$

where $E_{base}^{res}$ and $E_{tar}$ respectively represent the encoded feature segments specific to the $y_{base}$ and $y_{tar}$ portions. Based on the dual paths, the contrastive alignment process first extracts the attention features $\mathcal{F}_{ref}$ and $\mathcal{F}_{res}$ from the reference path and the response path. These features are derived from a variant of the standard attention mechanism, i.e. Attention $(K, Q, Q)$. They represent the response of the UNet features $Q_{ref}$ and $Q_{res}$ to the basic textual features $E_{base}$, where $Q_{ref}$ and $Q_{res}$ are features from the corresponding UNet cross-attention layer on the contrastive paths. By

constraining the similarity between the variant attention features $\mathcal{F}_{ref}$ and $\mathcal{F}_{res}$ from each cross-attention layer, SPF-Portrait encourages the representation of the base text in the response path to approach the behavior of the original model as:

$$
\begin{cases}
\mathcal{F}_{ref} = \text{Softmax}(\frac{K_{ref}(E_{base}^{ref})\ Q_{ref}^T}{\sqrt{d}})Q_{ref}, \\
\mathcal{F}_{res} = \text{Softmax}(\frac{K_{res}(E_{base}^{res})\ Q_{res}^T}{\sqrt{d}})Q_{res}, \\
\mathcal{L}_{\text{text-consistent}} = \sum_{j=1}^{L} \left\| \mathcal{F}_{ref}^j - \mathcal{F}_{res}^j \right\|_2^2,
\end{cases}
\tag{3}
$$

where $K_{ref}$ and $K_{res}$ denote the keys of $E_{base}^{ref}$ and $E_{base}^{res}$ in the dual paths. $L$ is the number of the attention layer in the denoising model. To improve consistency in fine-grained content, we further constrain the UNet features $Q$ from contrastive paths, which contain comprehensive information on portrait details and structures, e.g., posture, layout, etc. (Chung et al., 2024; Mo et al., 2024):

$$
\mathcal{L}_{\text{fine-grained}} = \sum_{j=1}^{L} \left\| Q_{ref}^j - Q_{res}^j \right\|_2^2.
\tag{4}
$$

### 3.3 SEMANTIC-AWARE FINE CONTROL MAP

Although such a dual-path contrastive approach can align customized portraits with original behavior both locally and globally, this vanilla alignment of intermediate features is too coarse and lacks precise control over the regions where alignment is applied. As shown in Fig. 4, it tends to suppress the responsiveness of target semantics, leading to the failure of target attribute customization. To address this issue, we propose a Semantic-Aware Fine Control Map (SFCM) that spatially guides the alignment process within appropriate regions, ensuring the effective response of target semantics. Specifically, during contrastive alignment, the spatial difference in noise predictions between dual paths can be used to serve as a prior knowledge for target response, forming a soft map $\mathcal{M}$:

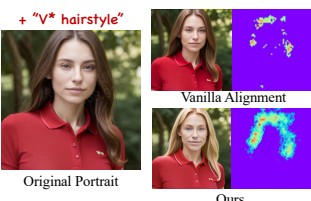

Figure 4: **The Motivation of SFCM.** The vanilla alignment results in the over-alignment of the customized portrait with the original portrait.

$$
\mathcal{M} = |\epsilon_\theta(x_t, t, E_{base}^{ref}) - \epsilon_{\theta'}(x_t, t, [E_{base}^{res}, E_{tar}])|,
\tag{5}
$$

where $\epsilon_{\theta'}$ and $\epsilon_\theta$ represent the noise prediction in the response and reference paths, respectively, $\theta'$ denotes the learnable parameters of the model in the response path. $[E_{base}^{res}, E_{tar}]$ represents complete textual feature input of the response path. $|\cdot|$ denotes element-wise absolute value. As analyzed in Fig. 2, due to semantic pollution causing the target response regions to diffuse into areas of unrelated attributes, the noise difference $\mathcal{M}$ is unable to accurately characterize the target response regions. Inspired by the insight that a phrase in the base text exhibits the lower semantic relevance to the target text, the regions highlighted by this phrase in $\mathcal{M}$ represent the higher degree of semantic pollution and should be excluded from $\mathcal{M}$. Based on this principle, we design the Semantic-Aware process to refine the soft map, which essentially involves progressively eliminating semantic pollution. For the input base text portion in the response path, we split it into multiple phrases, as shown in "Phrase embedding Attention Map" of Fig. 3. Concretely, for each textural feature of the phrase $E_{base}^{res}[i]$, $i = \{1, 2, \cdots, P\}$ ($P$ is the number of phrases in base text), we compute its mean of the cross-attention maps across all the denoising UNet layers to localize highlighted regions $\overline{A}_{base}[i]$ as:

$$
\overline{A}_{base}[i] = \frac{1}{L} \sum_{j=1}^{L} (A_{base}^j[i]),
\tag{6}
$$

where $A_{base}^j[i] = \text{AttentionMap}(E_{base}^{res}[i], j)$ represents the attention map of the $i$-th phrase embedding $E_{base}^{res}[i]$ at the $j$-th layer. Subsequently, to quantify the degree of exclusion, we leverage the representation capabilities of CLIP to calculate the similarity between $E_{tar}$ and each $E_{base}^{res}[i]$. We then weigh the $\overline{A}_{base}[i]$ based on the similarity and use it to refine the soft map $\mathcal{M}$:

$$
\widehat{\mathcal{M}} = \mathcal{M} - \sum_{i=1}^{P} \overline{A}_{base}[i] \cdot (1 - \gamma(i)),
$$
$$
\gamma(i) = D_{CLIP}(E_{base}^{res}[i], E_{tar}),
\tag{7}
$$

where $D_{CLIP}$ represents the cosine similarity in the CLIP embedding space. All attention maps $\overline{A}_{base}[i]$ are upsampled at a resolution of $64 \times 64$, i.e. the same size as soft map $\mathcal{M}$. $\widehat{\mathcal{M}}$ is the final map of our SFCM, as shown in Fig. 3 and Fig. 4, it represents the desired target response regions and can effectively prevent over-alignment with original portrait by precisely controlling the alignment process. Therefore, the text-consistent and fine-grained alignment constraints in Eq. 3 and Eq. 4 can be modified as follows:

$$\mathcal{L}_{M-\text{text}} = \sum_{j=1}^{L} \left\| (\mathcal{F}_{ref}^j - \mathcal{F}_{res}^j) \odot (1 - \widehat{\mathcal{M}}) \right\|_2^2,$$
$$\mathcal{L}_{M-\text{fine}} = \sum_{j=1}^{L} \left\| (Q_{ref}^j - Q_{res}^j) \odot (1 - \widehat{\mathcal{M}}) \right\|_2^2, \tag{8}$$

where $\odot$ denotes the hadamard product.

### 3.4 RESPONSE ENHANCEMENT VIA REPRESENTATION BIAS

In text-to-portrait customization, an excellent response to the target semantics and high-quality portrait output are essential. To reinforce the fidelity of the target attribute, previous works (Avrahami et al., 2022; Kim et al., 2022) directly apply cross-modal text-image supervision in CLIP space by employing the target text to supervise the one-step prediction (Yin et al., 2024), formulated as:

$$\mathcal{L}_{clip} = 1 - D_{CLIP}(\tau_{vision}(\hat{x}_{t\to0}) - \tau_{text}(E_{tar})),$$
$$\hat{x}_{t\to0} = \frac{\hat{x}_t}{\sqrt{\bar{\alpha}_t}} - \frac{\sqrt{1 - \bar{\alpha}_t}\epsilon_\theta(\hat{x}_t, t, \tau_{text}([E_{base}^{res}, E_{tar}]))}{\sqrt{\bar{\alpha}_t}}, \tag{9}$$

where $\tau_{vision}$ and $\tau_{text}$ denote the CLIP vision and text encoders, respectively, and $\hat{x}_{t\to0}$ denotes the one-step prediction of $x_t$ in the $t$-th timestep.

However, direct text-image supervision in Eq. 9 overlooks the cross-modal representation gap between text and image, causing the model to overfit the textual description during optimization. As illustrated in Fig. 5, it ultimately leads to degradation of the visual quality of the generated portraits. To address this issue, we propose a novel target response enhancement mechanism that achieves an effect similar to supervision within the same modality, improving the fidelity of the target attribute while ensuring high-quality portrait output. Specifically, we propose a representation bias $\Delta(\cdot, \cdot)$, representing the difference between the representations of the CLIP textual space and the CLIP visual space (Abdelfattah et al., 2023; Xue et al., 2022). By introducing the ground truth image $x_0$ (an arbitrary image with the target attribute), we obtain the representation bias between target text and ground truth image $x_0$, i.e. $\Delta(x_0, E_{tar})$, as well as that between target text and one-step prediction $\hat{x}_{t\to0}$, i.e. $\Delta(\hat{x}_{t\to0}, E_{tar})$, formulated as:

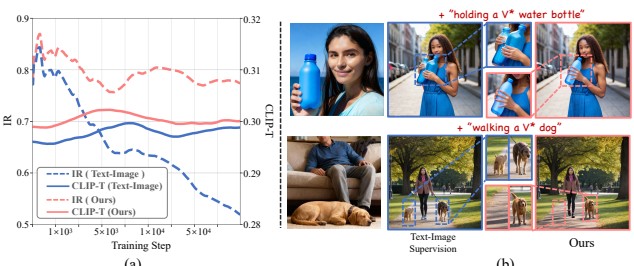

Figure 5: **Comparison with Direct Text-Image Supervision. (a)** illustrates the Image-Reward (IR) and CLIP Text-Alignment Score (CLIP-T) across training steps. Image-Reward (Xu et al., 2023) is a metric used to evaluate image quality. **(b)** displays samples from the target attribute fine-tuning dataset, direct text-image supervision (Avrahami et al., 2022), and our method.

$$\Delta(x_0, E_{tar}) = \tau_{vision}(x_0) - \tau_{text}(E_{tar}),$$
$$\Delta(\hat{x}_{t\to0}, E_{tar}) = \tau_{vision}(\hat{x}_{t\to0}) - \tau_{text}(E_{tar}), \tag{10}$$

Subsequently, we constrain the similarity of these two representation biases:

$$\mathcal{L}_{enhanced} = 1 - D_{CLIP}(\Delta(\hat{x}_{t\to0}, E_{tar}), \Delta(x_0, E_{tar})). \tag{11}$$

As shown in Fig. 5, this loss simultaneously enhances the performance of the target semantics while ensuring the high quality of the portrait output. Finally, the overall optimization objective can be represented as:

$$\mathcal{L}_{SPF} = \mathcal{L}_{diff} + \underbrace{\lambda_1 \mathcal{L}_{M-\text{text}} + \lambda_2 \mathcal{L}_{M-\text{fine}}}_{\text{alignment}} + \underbrace{\lambda_3 \mathcal{L}_{enhacned}}_{\text{response}}, \tag{12}$$

where $\lambda_1$, $\lambda_2$ and $\lambda_3$ are the hyperparameters.

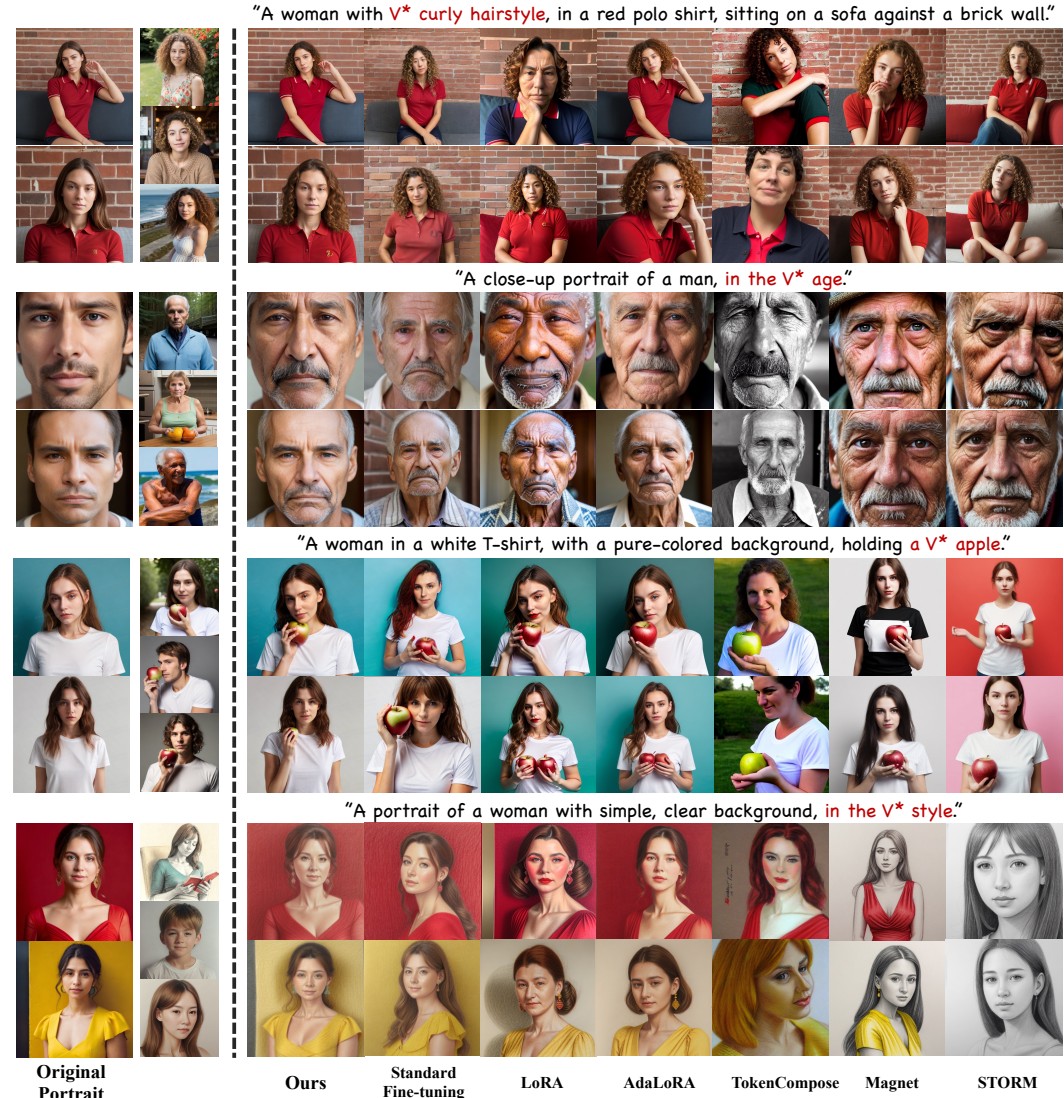

Figure 6: **Qualitative Comparisons between Our Method and SOTA Methods** in terms of diverse personalized attributes such as hairstyle, age, and image style. For each target attribute, we evaluate two cases using different random seeds. More results are provided in Appendix Fig. 16 and Fig. 19.

# 4 EXPERIMENTS

## 4.1 EXPERIMENTAL SETUP

**Implementation Details.** We adopt the SD1.5 (Rombach et al., 2022) and SD3.5-M (Esser et al., 2024) as our base model. The hyperparameters $\lambda_1$, $\lambda_2$ and $\lambda_3$ are set to 0.2, 0.1 and 0.6. More details about the experiments are provided in the *Appendix*.

**Dataset.** Our training set contains 230K diverse portraits with new attributes (e.g., skin textures, hairstyles), captioned by GPT-4o (Achiam et al., 2023) and Cambrian-1 (Tong et al., 2024). For evaluation, we create a test set of 5K triples, each with: (1) an original caption, (2) its corresponding original portrait, and (3) a target caption of customized attributes. More details are in the *Appendix*.

**Evaluation Metrics.** We evaluate three key aspects: (1) preservation of the original model's behavior, (2) responsiveness to target semantics, and (3) overall image quality. Concretely, we employ FID (Heusel et al., 2017), LPIPS (Zhang et al., 2018), identity similarity (ID), CLIP Image Score (CLIP-I) (Radford et al., 2021), and segmentation consistency (Kirillov et al., 2023) (Seg-Cons) to measure the consistency between the original and customized portraits. We use the CLIP Text-Alignment Score (CLIP-T) (Radford et al., 2021) to evaluate the responsiveness to target semantics. For overall image quality assessment, we use HPSv2 (Wu et al., 2023) and MPS (Zhang et al., 2024).

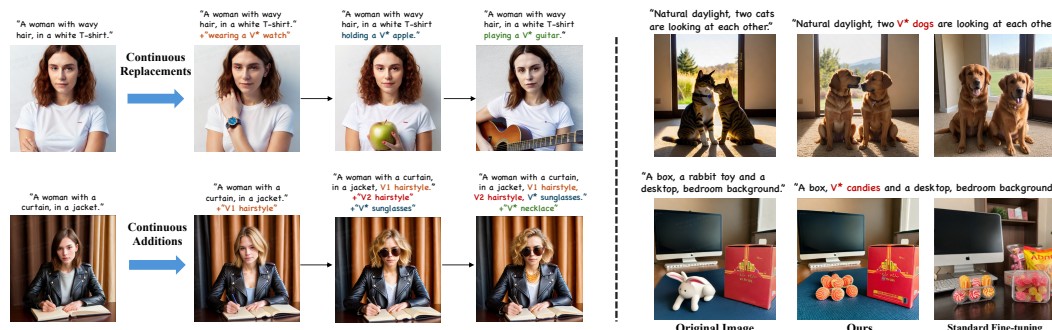

(a) Continuous Customization  (b) Extension to General T2I Customization

Figure 7: **(a)** Results of continuous replacements and additions of target semantics in pure text-to-portrait customization. More results are shown in Appendix Fig. 17. **(b)** Results of extending our method to achieve general pure T2I customization. More results are provided in Appendix Fig. 18.

## 4.2 QUALITATIVE EVALUATION

**Comparison with SOTAs.** We qualitatively compare our method with the SOTA approaches, including standard fine-tuning (Rombach et al., 2022), PEFT-based methods such as LoRA (Hu et al., 2021) and AdaLoRA (Zhang et al., 2023b), and decoupled methods such as Tokencompose (Wang et al., 2024c), Magenet (Zhuang et al., 2024), and STORM (Han et al., 2025). As shown in Fig. 6, although LoRA and AdaLoRA tend to retain the original behavior in some cases, their performance is unstable in detail alignment. For instance, in the customization of "V* age", LoRA causes a noticeable change in identity in both cases, while in the second case of hairstyle customization, AdaLoRA completely transforms the portrait posture. Magnet, TokenCompose and STORM naively follow the input text conditions entirely, ignoring the preservation of the original model's behavior across all test cases. For example, the customization of "V* style" results in a complete alteration of the portrait in all cases. In contrast, our method purely customizes the target attributes while preserving the original model's behavior in aspects such as background, pose, and identity. It demonstrates that our approach effectively addresses semantic pollution during portrait customization.

**Continuous Customization.** Based on real-world application scenarios, we conducted a qualitative evaluation of our method in continuous portrait customization task. As shown in Fig. 7 (a), our method reliably excels in the continuous replacement and addition of target semantics, which effectively customizes portrait attributes while mitigating interference with original behavior. It indicates that SPF-Portrait has the potential to play a role in the application of continuous AI portrait creation.

**Extension to General T2I Customization.** Although in this paper, we primarily focus on pure text-to-portrait customization due to its application value and the challenging need for more precise attribute customization. However, as shown in Fig. 7 (b), we attempt to apply SPF-Portrait to General T2I Customization. The excellent experimental results demonstrate the feasibility of extending our method to address the issue of semantic pollution in general T2I customization.

Table 1: **Quantitative Comparison Results.** In our specific pairwise comparison, unlike general image generation, lower FID values reflect greater consistency with the original model's behavior. Notably, the underlined values in "Ours (w/o SFCM)" are unusually low because the generated portraits may overly align with the original portraits. The results of the ablation study are based on SD1.5, while the SD3.5-based results are provided in the *Appendix*.

| Method | Preservation | | | | | Responsiveness | Overall | |
|---|---|---|---|---|---|---|---|---|
| | FID (↓) | LPIPS (↓) | ID (↑) | CLIP-I (↑) | Seg-Cons (↑) | CLIP-T (↑) | HPSv2 (↑) | MPS(↑) |
| Standard Fine-tuning (Rombach et al., 2022) | 20.41 | 0.57 | 0.21 | 0.63 | 57.77 | 0.24 | 0.21 | 0.67 |
| LoRA (Hu et al., 2021) | 9.82 | 0.38 | 0.52 | 0.71 | 58.37 | 0.27 | 0.23 | 1.21 |
| AdaLoRA (Zhang et al., 2023b) | 7.38 | 0.40 | 0.39 | 0.80 | 64.86 | 0.23 | 0.24 | 1.10 |
| TokenCompose (Wang et al., 2024c) | 10.93 | 0.41 | 0.32 | 0.81 | 40.22 | 0.27 | 0.24 | 0.71 |
| Magnet (Zhuang et al., 2024) | 18.92 | 0.48 | 0.38 | 0.61 | 32.87 | 0.26 | 0.26 | 0.97 |
| STORM (Han et al., 2025) | 17.30 | 0.54 | 0.27 | 0.60 | 30.04 | 0.26 | 0.24 | 0.70 |
| **Ours (SD 3.5-M)** | **4.27** | **0.30** | **0.65** | **0.82** | **77.18** | **0.32** | **0.30** | **1.83** |
| **Ours (SD 1.5)** | **4.50** | **0.35** | **0.55** | **0.83** | **75.74** | **0.30** | **0.28** | **1.49** |
| Ours (w/o $\mathcal{L}_{M-text}$) | 4.97 | 0.39 | 0.48 | 0.60 | 61.39 | 0.28 | 0.23 | 1.13 |
| Ours (w/o $\mathcal{L}_{M-fine}$) | 6.74 | 0.42 | 0.32 | 0.71 | 41.62 | 0.27 | 0.21 | 1.22 |
| Ours (w/o $\mathcal{L}_{enhanced}$) | 4.52 | 0.37 | 0.49 | 0.81 | 74.38 | 0.22 | 0.23 | 1.40 |
| Ours (w/o SFCM) | _4.13_ | _0.14_ | _0.73_ | _0.88_ | _80.03_ | 0.17 | 0.23 | 1.09 |

holding a V* toy bear

"A Boy, in a white T-shirt, with gray background, holding a V* toy bear."

Original Portrait    Standard Fine-tuning    w/o $L_{M-text}$    w/o $L_{M-fine}$    w/o $L_{enhanced}$    w/o SFCM    Ours

Figure 9: **Qualitative Ablation Study.** We independently ablate the losses and the SFCM module.

### 4.3 QUANTITATIVE EVALUATION

**Metric Evaluation.** Tab. 1 shows the quantitative results of our method against baselines on the test set. Both our method based on SD1.5 (Rombach et al., 2022) and SD3.5-M (Esser et al., 2024) show substantial improvement in preserving the original behavior compared to all competitors, achieving state-of-the-art performance across all metrics. It is notable that our method significantly outperforms competitors in "Seg-Cons", demonstrating pixel-level alignment precision that far surpasses other approaches. The optimal CLIP-T and overall scores confirm that our method enhances the response to target semantics and achieves higher-quality portrait customization.

**User Study.** We conduct a user study to have a comprehensive assessment of our method. We design three dimensions for evaluation: Original Behavior Consistency (OBC), Target Attribute Responsiveness (TAR), and Aesthetic Preference (AP). We invite 32 participants from different social backgrounds, with each test session lasting about 30 minutes. Users perform pairwise comparisons between our method and competitors in three dimensions. The results are as shown in Fig. 8, our method defeats all competitors in all dimensions, especially in

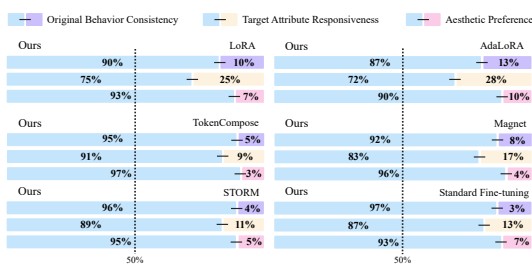

Figure 8: **User Study.** The percentages show the proportion of users who select the corresponding method.

OBC and TAR. This highlights our powerful ability in preserving the original model's behavior while purely adapting to new attributes. Please refer to the *Appendix* for more details.

### 4.4 ABLATION STUDY

To validate the effectiveness of different components of our method, we conduct thorough ablation studies. Qualitative results, shown in Fig. 9, indicate that the absence of $\mathcal{L}_{M-text}$ results in weaker alignment of base text response with the original portrait, while the lack of $\mathcal{L}_{M-fine}$ leads to inconsistencies in detailed content, such as portrait posture. Without $\mathcal{L}_{enhanced}$, the fidelity of the target attribute significantly degrades, failing to follow the action of 'holding' and with a tendency to disrupt the spatial coherence of the "V* toy bear". Superior performance across all metrics of quantitative results in the ablation part of Tab. 1 further confirm these visual observations. Notably, "w/o SFCM" shows superior preservation metrics in Table 1. However, as shown in Figure 9, it disregards the target semantics and over-aligns the target attribute with the original portrait. Such outcomes represent a failure that should not be overlooked in our task.

### 5 CONCLUSION

In this paper, we propose SPF-Portrait, a novel fine-tuning framework designed to address the issue of Semantic Pollution for pure text-to-portrait customization. By introducing the original model as a reference path and utilizing contrastive alignment, we achieve the goals of purely achieving the customized semantics. We precisely retain the original model's behavior and ensure an effective response to target semantics by innovatively designing a Semantic-Aware Fine-Control Map to guide the alignment process and a response enhancement mechanism for target semantics. Extensive experiments show that our method can achieve the SOTA performance. However, we acknowledge that our SFCM approach, which relies on attention maps, is less effective with abstract descriptions like "the background is softly blurred." This limitation underscores the need for more generalizable mechanisms to preserve model behavior across varied text descriptions.

ETHICS STATEMENT

This work adheres to the ICLR Code of Ethics. In this study, no animal experimentation was involved. All datasets used, including Pick-a-pic and our collecting video preference dataset, were sourced in compliance with relevant usage guidelines, ensuring no violation of privacy. We have taken care to avoid any biases or discriminatory outcomes in our research process. No personally identifiable information was used, and no experiments were conducted that could raise privacy or security concerns. We are committed to maintaining transparency and integrity throughout the research process.

REPRODUCIBILITY STATEMENT

To ensure reproducibility, we have made the following efforts: (1) We will release our code and the collected dataset. (2) We provide experiments setup in Sec. 4 and the more details about training process are presented in Appendix. C including training steps, model configurations, and hardware details. (3) We elaborate on our evaluation protocol in detail in Sec. 4. We believe these measures will enable other researchers to reproduce our work and further advance the field.

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

# APPENDIX

## A   OVERVIEW OF THE APPENDIX

- The explanation of mathematical notations in SPF-Portrait is provided in Tab. 2.
- The use of large language models (LLMs) is provided in Sec. B.
- The implementation details of the SPF-Portrait are provided in Sec. C.
- DiT-based implementation of the SPF-Portrait is provided in Sec. D.
- The analysis of the fine-tuning architecture is provided in Sec. E.
- The analysis of the fine-tuned model is provided in Sec. F.
- The sensitivity analysis of loss is provided in Sec. G.
- The ablation study of the training stage is provided in Sec. H.
- Details of the user study are provided in Sec. J.
- More experimental results are shown in Fig. 16, Fig. 17 and Fig. 18.

Table 2: Mathematical Notations in SPF-Portrait

| Symbol | Description |
|---|---|
| $x_0$ | Original clean image sample in diffusion process |
| $x_{1:T}$ | Noisy samples generated through forward diffusion process over T steps |
| $\epsilon$ | Gaussian noise added during diffusion process |
| $\epsilon_\theta$ | Denoising model with learnable parameters $\theta$ |
| $t$ | Diffusion timestep, $t \in [0, T]$ |
| $y$ | Textual conditions/input prompts |
| $E$ | Textual features, $E = \tau_{\text{text}}(y)$ |
| $\tau_{\text{text}}$ | Text encoder |
| $y_{\text{base}}$ | Base text prompt (original description) |
| $y_{\text{tar}}$ | Target text prompt (customized attributes) |
| $E_{\text{base}}^{\text{ref}}$ | Reference path's base text features |
| $E_{\text{base}}^{\text{res}}$ | Response path's base text features |
| $E_{\text{tar}}$ | Target text features |
| $\mathcal{F}_{\text{ref}}$ | Attention features from reference path |
| $\mathcal{F}_{\text{res}}$ | Attention features from response path |
| $Q_{\text{ref}}$ | UNet features from reference path's cross-attention layer |
| $Q_{\text{res}}$ | UNet features from response path's cross-attention layer |
| $K_{\text{ref}}$ | Key for $E_{\text{base}}^{\text{ref}}$ in reference path |
| $K_{\text{res}}$ | Key for $E_{\text{base}}^{\text{res}}$ in response path |
| $d$ | Dimension scaling factor in attention computation |
| $L$ | Number of attention layers in denoising model |
| $\mathcal{M}$ | Soft map of spatial difference in noise predictions |
| $\widehat{\mathcal{M}}$ | Final Semantic-Aware Fine Control Map (SFCM) |
| $\bar{A}_{\text{base}}[i]$ | Mean attention map for i-th phrase in base text |
| $\gamma(i)$ | CLIP similarity score between phrases |
| $\Delta(\cdot, \cdot)$ | Difference vector between CLIP visual/text spaces |
| $\hat{x}_{t \rightarrow 0}$ | One-step prediction in diffusion process |
| $\alpha_t, \bar{\alpha}_t$ | Noise schedule parameters in diffusion process |
| $\lambda_1, \lambda_2, \lambda_3$ | Loss weighting hyperparameters |
| $\odot$ | Hadamard (element-wise) product operator |

## B THE USE OF LARGE LANGUAGE MODELS (LLMS)

Large Language Models (LLMs) were used to aid or polish the writing of this manuscript. Specifically, we used Claude-4-Sonnet solely for language polishing and grammatical refinement of the written text. All research contributions, including the main ideas, technical approaches, experimental work, and scientific insights presented in this paper, are entirely the work of the human authors. The LLM usage was limited to improving the clarity and readability of the already-written content without altering the substance or meaning of our work.

## C IMPLEMENTATION DETAILS OF SPF-PORTRAIT

### C.1 DETAILS OF TRAINING

As shown in Fig. 10, the training process of our approach consists of two stages: fine-tuning with all the parameters updated in the first stage and contrastive alignment learning in the second stage. In the first stage, we employ standard fine-tuning (Rombach et al., 2022) to learn target attributes, which aims to enable the T2I model to rapidly adapt to the target concepts of our dataset. For this stage, we train the model using 8 V100 GPUs with a batch size of 8, iterating for 2 epochs and a learning rate of 1e-5. In the second stage, the training process follows the approach outlined in the main text. The goal is to enable the T2I model to grasp pure target concepts without compromising the original model's performance, thereby preventing semantic pollution caused by the target text. Due to the additional memory consumption of the dual-branch architecture, we set the batch size to 2, iterating for 5 epochs with a learning rate of 5e-5. We summarize the complete training procedure of TAPO in Algorithm. 1. The same dataset and optimizer (AdamW with default parameters: beta1=0.9, beta2=0.999, weight decay=0.01) are used for both the first and second stages.

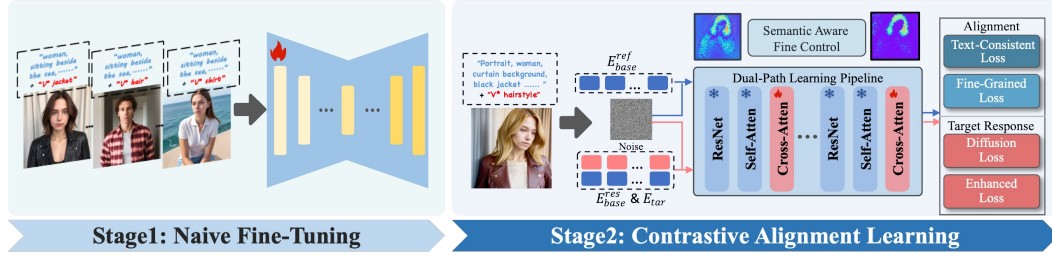

Figure 10: Our overall training pipeline.

Due to the dual-path training framework in second stage, which requires an additional frozen original model compared to standard fine-tuning, our approach incurs extra memory costs and increased computation time. We provide the corresponding resource consumption details in Tab. 3. The frozen model (which doesn't participate parameter updates) adds only 5GB of GPU memory overhead under typical FP32 precision settings.

Table 3: **Computation Time and Memory Usage of Training under Different Data Types.** The data in bold represents our implementation configuration.

| Method | Data Type | | |
| --- | --- | --- | --- |
| | FP16 | FP32 | BF16 |
| Stage-1 (w/o reference path) | 1.92s/iter (17GB) | **2.28s/iter (23GB)** | 1.90s/iter (18GB) |
| Stage-2 (w/ reference path) | 2.10s/iter (21GB) | **3.26s/iter (28GB)** | 2.15s/iter (18GB) |
| Standard Fine-tuning (Rombach et al., 2022) | 1.92s/iter (17GB) | **2.28s/iter (23GB)** | 1.90s/iter (18GB) |

**Algorithm 1** SPF-Portrait Training Procedure

1: **Input:** Pre-trained T2I model $\theta_0$, portrait dataset $\mathcal{D}$, base text $y_{\text{base}}$, target text $y_{\text{tar}}$
2: **Hyperparameters:** $\lambda_1, \lambda_2, \lambda_3$
3: **Output:** Fine-tuned model $\theta'$
4: **procedure** STAGE1: STANDARD FINE-TUNING
5:   Initialize $\theta' \leftarrow \theta_0$
6:   **for** each training step **do**
7:    Sample $(x_0, y_{\text{base}}, y_{\text{tar}}) \sim \mathcal{D}$
8:    Encode text: $E \leftarrow \tau_{\text{text}}([y_{\text{base}}, y_{\text{tar}}])$
9:    Sample $t \sim \mathcal{U}(0, T), \epsilon \sim \mathcal{N}(0, I)$
10:    Compute $x_t \leftarrow \sqrt{\bar{\alpha}_t} x_0 + \sqrt{1 - \bar{\alpha}_t} \epsilon$
11:    Predict noise: $\hat{\epsilon} \leftarrow \epsilon_{\theta'}(x_t, t, E)$
12:    Update $\theta'$ using $\mathcal{L}_{\text{diff}} = \|\epsilon - \hat{\epsilon}\|_2^2$ Eq. 1
13:   **end for**
14: **end procedure**
15: **procedure** STAGE2: DUAL-PATH CONTRASTIVE LEARNING
16:   Freeze reference model: $\theta \leftarrow \theta_0$
17:   Initialize response model from Stage1
18:   **for** each training step **do**
19:    Sample $(x_0, y_{\text{base}}, y_{\text{tar}}) \sim \mathcal{D}$
20:    Sample $t \sim \mathcal{U}(0, T), \epsilon \sim \mathcal{N}(0, I)$
21:    Compute $x_t \leftarrow \sqrt{\bar{\alpha}_t} x_0 + \sqrt{1 - \bar{\alpha}_t} \epsilon$
22:    $E_{\text{base}}^{\text{ref}} \leftarrow \tau_{\text{text}}(y_{\text{base}}), \quad \epsilon_{\text{ref}} \leftarrow \epsilon_\theta(x_t, t, E_{\text{base}}^{\text{ref}})$ (Eq. 2)    ▷ *# Reference path (frozen)*
23:    $E_{\text{base}}^{\text{res}}, E_{\text{tar}} \leftarrow \tau_{\text{text}}([y_{\text{base}}, y_{\text{tar}}]), \epsilon_{\text{res}} \leftarrow \epsilon_{\theta'}(x_t, t, [E_{\text{base}}^{\text{res}}, E_{\text{tar}}])$ (Eq. 2)   ▷ *# Response path*
  *(trainable)*
24:    *# Compute SFCM*
25:    $\mathcal{M} \leftarrow |\epsilon_{\text{ref}} - \epsilon_{\text{res}}|$ (Eq. 5)
26:    Split $E_{\text{base}}^{\text{res}}$ into phrases $[E_1, ..., E_P]$
27:    **for** $i = 1$ to $P$ **do**
28:     $\bar{A}_{\text{base}}[i] \leftarrow \frac{1}{L} \sum_{j=1}^{L} A_{\text{base}}^j[i]$ (Eq. 6)
29:     $\gamma(i) \leftarrow D_{\text{CLIP}}(E_i, E_{\text{tar}})$
30:    **end for**
31:    $\widehat{\mathcal{M}} \leftarrow \mathcal{M} - \sum_{i=1}^{P} \bar{A}_{\text{base}}[i] \cdot (1 - \gamma(i))$ (Eq. 7)
32:    *# Feature extraction and alignment*
33:    **for** each cross-attention layer $j$ **do**
34:     Extract $\mathcal{F}_{\text{ref}}^j, \mathcal{F}_{\text{res}}^j, Q_{\text{ref}}^j, Q_{\text{res}}^j$
35:     $\mathcal{L}_{M-\text{text}} \leftarrow \mathcal{L}_{M-\text{text}} + \|(\mathcal{F}_{\text{ref}}^j - \mathcal{F}_{\text{res}}^j) \odot (1 - \widehat{\mathcal{M}})\|_2$ (Eq. 8)
36:     $\mathcal{L}_{M-\text{fine}} \leftarrow \mathcal{L}_{M-\text{fine}} + \|(Q_{\text{ref}}^j - Q_{\text{res}}^j) \odot (1 - \widehat{\mathcal{M}})\|_2$ (Eq. 8)
37:    **end for**
38:    *# Response enhancement*
39:    $\hat{x}_0 \leftarrow \frac{\hat{x}_t}{\sqrt{\bar{\alpha}_t}} - \frac{\sqrt{1 - \bar{\alpha}_t} \epsilon_\theta(\hat{x}_t, t, \tau_{\text{text}}([E_{\text{base}}^{\text{res}}, E_{\text{tar}}]))}{\sqrt{\bar{\alpha}_t}}$ (Eq. 9)
40:    $\Delta(\hat{x}_{t \to 0}, E_{\text{tar}}) \leftarrow \tau_{\text{vision}}(\hat{x}_{t \to 0}) - \tau_{\text{text}}(E_{\text{tar}}), \quad \Delta(x_0, E_{\text{tar}}) \leftarrow \tau_{\text{vision}}(x_0) - \tau_{\text{text}}(E_{\text{tar}})$
41:    $\mathcal{L}_{\text{enhanced}} \leftarrow 1 - D_{\text{CLIP}}(\Delta(\hat{x}_{t \to 0}, E_{\text{tar}}), \Delta(x_0, E_{\text{tar}}))$ (Eq. 11)
42:    *# Total loss and update*
43:    $\mathcal{L}_{\text{SPF}} \leftarrow \mathcal{L}_{\text{diff}} + \lambda_1 \mathcal{L}_{M-\text{text}} + \lambda_2 \mathcal{L}_{M-\text{fine}} + \lambda_3 \mathcal{L}_{\text{enhanced}}$ (Eq. 12)
44:    Update $\theta'$ using $\nabla_{\theta'} \mathcal{L}_{\text{SPF}}$
45:   **end for**
46: **end procedure**

## C.2 DETAILS OF TRAINING DATASET

Our work focuses on preventing semantic pollution in fine-tuning portrait T2I models while enabling the model to learn the concepts from the target attributes. To achieve this, we constructed a dataset containing various image-text pairs related to portrait concepts for training the T2I diffusion model.

Considering the quality and diversity of the dataset, we utilized widely adopted community (civiti) checkpoints for portrait generation as the checkpoints for the Stable Diffusion (SD) model, including RealVisXL_V1.0 and HumanModel, to generate portrait images encompassing a wide range of attributes. The attribute statistics and corresponding samples are shown in Fig. 11, respectively.

To improve dataset quality, we focus on two aspects: 1) enhancing image-text alignment using FLIP (Li et al., 2024), a CLIP checkpoint specifically for portraits, to retain the top 30% of matching pairs, and 2) improving visual fidelity by filtering images with a Human Aesthetic Preference Score (HPS) and Image-Reward (IR).

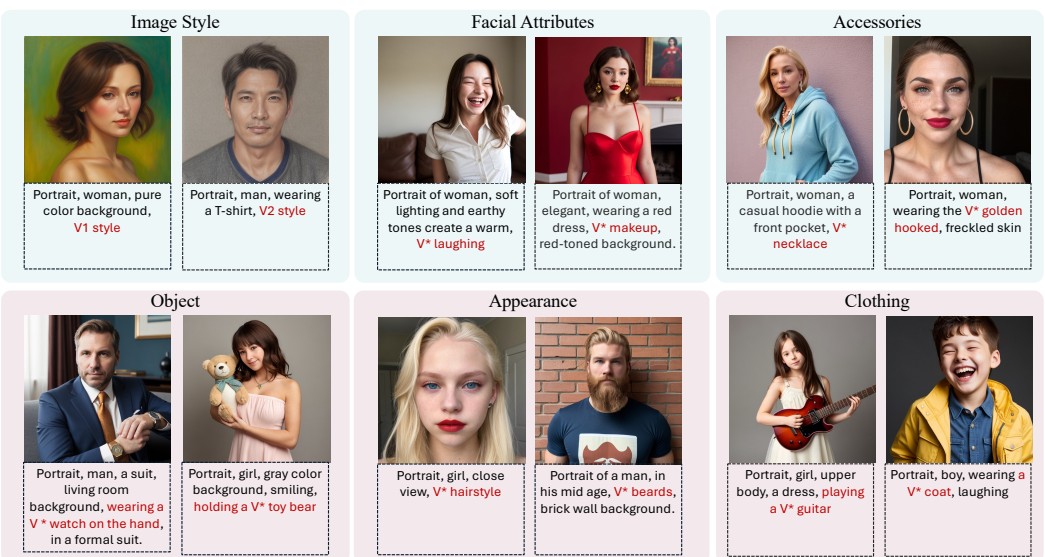

Figure 11: **Examples of our training datasets.**

## D DiT-BASED IMPLEMENTATION OF SPF-PORTRAIT

To demonstrate the generalization of our SPF-Portrait on advanced diffusion architectures, we conduct experiments using SD3.5-M (Esser et al., 2024) as the base model. SD3.5-M utilized the DiT (Peebles & Xie, 2023) as the backbone and employs rectified flow to reformulate the denoising process as an ordinary differential equation. The core of these methods is to train a neural network $v_{t,\theta}$ to satisfy the velocity field by minimizing the Flow Matching objective:

$$\mathcal{L}_{RF-diff} = \mathbb{E}_{t\in[0,1],\boldsymbol{x}_t \sim p_t} \|\boldsymbol{v}_t(\boldsymbol{x}_t) - \boldsymbol{v}_\theta(\boldsymbol{x}_t)\|^2. \tag{13}$$

where $x_t = (1-t)x_0 + tx_1$. The main modifications for DiT involve two aspects: 1) Attention feature enhancement: MM-DiT blocks use self-attention on concatenated image-text features instead of cross-attention interaction. 2) Attention map collection: Since SD3.5 introduces the additional T5 text encoder, phrase attention map computation must consider attention maps from both encoders.

**Dual-Path.** In the second stage, our dual-path architecture comprises two branches: 1) Reference Path – it loads the original pre-trained model (now a DiT-based diffusion model) and takes only the base text embedding $E_{base}^{ref}$ as input, using its intermediate representations to anchor the original model's behavior; 2) Response Path – it is initialized from the stage-one standard fine-tuned model (also converted to a DiT-based diffusion backbone) and accepts both base embeddings $E_{base}^{res}$ and target text embeddings $E_{tar}$.

**Alignment Process.** Specifically, in the MM-DiT block, we use the key and value from visual features as the $K^{vis}$ and $V_{vis}$, while using text feature query as $Q^{txt}$ The text consistent loss can be calculate as follow:

$$
\begin{cases}
\mathcal{F}_{ref} = \text{Softmax}(\frac{Q_{ref}^{txt}(E_{base}^{ref}) \ K_{ref}^{vis\ T}}{\sqrt{d}})V_{ref}^{vis}, \\
\mathcal{F}_{res} = \text{Softmax}(\frac{Q_{res}^{txt}(E_{base}^{res}) \ K_{res}^{vis\ T}}{\sqrt{d}})V_{res}^{vis}, \\
\mathcal{L}_{\text{text-consistent}} = \sum_{j=1}^{L} \left\| \mathcal{F}_{ref}^j - \mathcal{F}_{res}^j \right\|_2^2.
\end{cases}
\tag{14}
$$

where $L$ denotes the MM-DiT layer number here. To enhance consistency in fine-grained content, we further constrain the visual features $Q^{vis}$ from contrastive paths, and the fine-grained loss can be reformulated as:

$$
\mathcal{L}_{\text{fine-grained}} = \sum_{j=1}^{L} \left\| Q_{ref}^{vis,\ j} - Q_{res}^{vis,\ j} \right\|_2^2.
\tag{15}
$$

Since SD 3.5 builds upon Rectified Flow, we reformulate the noise difference in Eq. 5 to represent the velocity difference in our dual-path architecture:

$$
\mathcal{M} = |v_\theta(x_t, t, E_{base}^{ref}) - v_{\theta'}(x_t, t, [E_{base}^{res}, E_{tar}])|,
\tag{16}
$$

Regarding the Phrase Embedding Attention Map, we follow (Wei et al., 2024; Cai et al., 2025) by extracting attention maps of the corresponding tokens encoded by T5 and CLIP. The map can be calculated as:

$$
\overline{A}_{base}[i] = \frac{1}{2L} \sum_{j=1}^{L} (A_{base,\text{CLIP}}^j[i] + A_{base,\text{T5}}^j[i]).
\tag{17}
$$

We subsequently compute the SFCM following the same procedure as Eq. 7:

$$
\widehat{\mathcal{M}} = \mathcal{M} - \sum_{i=1}^{P} \overline{A}_{base}[i] \cdot (1 - \gamma(i)),
$$
$$
\gamma(i) = D_{CLIP}(E_{base}^{res}[i], E_{tar}),
\tag{18}
$$

Through the above modifications, we demonstrate that our framework is also compatible with the DiT architecture. The quantitative and ablation results are shown Tab. 4 and qualitative results are presented in Fig. 21.

Table 4: **Quantitative and Ablation Results.** In our specific pairwise comparison, unlike general image generation, lower FID values reflect greater consistency with the original model's behavior. Notably, the underlined values in "Ours (w/o SFCM)" are unusually low because the generated portraits may overly align with the original portraits.

| Method | Preservation | | | | | Responsiveness | Overall | |
|---|---|---|---|---|---|---|---|---|
| | FID ($\downarrow$) | LPIPS ($\downarrow$) | ID ($\uparrow$) | CLIP-I ($\uparrow$) | Seg-Cons ($\uparrow$) | CLIP-T ($\uparrow$) | HPSv2 ($\uparrow$) | MPS($\uparrow$) |
| Standard Fine-tuning (Rombach et al., 2022) | 20.41 | 0.57 | 0.21 | 0.63 | 57.77 | 0.24 | 0.21 | 0.67 |
| LoRA (Hu et al., 2021) | 9.82 | 0.38 | 0.52 | 0.71 | 58.37 | 0.27 | 0.23 | 1.21 |
| AdaLoRA (Zhang et al., 2023b) | 7.38 | 0.40 | 0.39 | 0.80 | 64.86 | 0.23 | 0.24 | 1.10 |
| TokenCompose (Wang et al., 2024c) | 10.93 | 0.41 | 0.32 | 0.81 | 40.22 | 0.27 | 0.24 | 0.71 |
| Magnet (Zhuang et al., 2024) | 18.92 | 0.48 | 0.38 | 0.61 | 32.87 | 0.26 | 0.26 | 0.97 |
| STORM (Han et al., 2025) | 17.30 | 0.54 | 0.27 | 0.60 | 30.04 | 0.26 | 0.24 | 0.70 |
| **Ours (SD3.5-M)** | **4.27** | **0.30** | **0.65** | **0.82** | **77.18** | **0.32** | **0.30** | **1.83** |
| Ours (w/o $\mathcal{L}_{M-text}$) | 5.24 | 0.41 | 0.39 | 0.67 | 53.24 | 0.25 | 0.25 | 1.10 |
| Ours (w/o $\mathcal{L}_{M-fine}$) | 7.72 | 0.48 | 0.27 | 0.54 | 31.94 | 0.22 | 0.25 | 1.18 |
| Ours (w/o SFCM) | 3.72 | 0.11 | 0.89 | 0.92 | 87.24 | 0.12 | 0.23 | 1.14 |

## E    ANALYSIS OF FINE-TUNING ARCHITECTURE

During the contrastive learning of the second stage, our approach exclusively trains the parameters in the cross-attention modules. We compare results across various network architectures, including

"full-weight", "LoRA on cross-attention", and "additional adapters". As illustrated in Fig. 12, all architectures under our contrastive learning achieve some level of alignment. Notably, "LoRA on cross-attention", "Adapter on Cross-Atten" and "Cross-Atten(ours)" outperform the "full weights" in alignment, this is because the diffusion model relies on the cross-attention mechanism for text-conditioned control, and optimizing the most critical parameters enables a better understanding of independent target attributes. However, "LoRA on Cross-Atten", due to its limited learnable parameters, falls short in understanding the original behavior compared to our method. Ours achieves a superior balance between alignment and attribute learning. "Adapter on Cross-Atten" achieves the suboptimal performance, as it independently adjusts all the parameters of cross-attention module. However, the isolated attention structure limits the interaction between target text features and base text features, rendering in partial misalignment. The results in Tab. 5 further validate our conclusions.

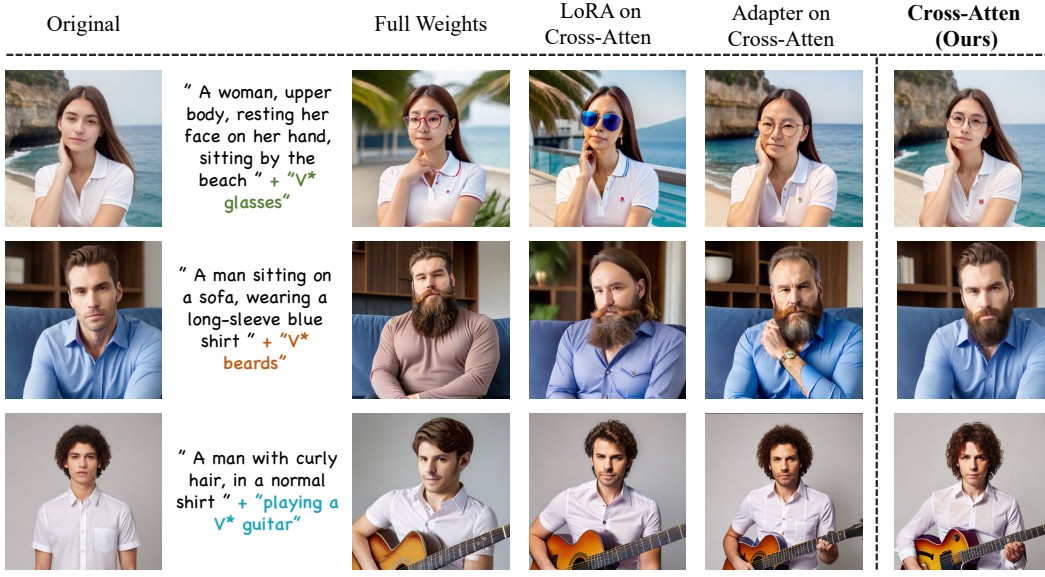

Figure 12: **Comparison of results across different updated network architectures in our contrastive pipeline.** "Full Weights" indicates that all network parameters are updated, "LoRA on Cross-Atten" refers to the integration of LoRA into the Cross-Attention modules, and "Adapter on Cross-Atten" denotes the addition of parallel cross-attention layers, akin to IP-adapter (Ye et al., 2023).

Table 5: **Quantitative Comparisons** with other architecture.

| Method | Preservation | | | | | Overall | | Responsiveness |
|---|---|---|---|---|---|---|---|---|
| | FID (↓) | LPIPS (↓) | ID (↑) | CLIP-I (↑) | Seg-Cons (↑) | HPSv2 (↑) | MPS(↑) | CLIP-T ( ↑) |
| Full Weights | 7.82 | 0.40 | 0.309 | 0.81 | 48.39 | 0.22 | 0.87 | 0.26 |
| LoRA on Cross-Atten | 7.10 | 0.39 | 0.487 | 0.61 | 68.37 | 0.24 | 1.21 | 0.26 |
| Adapter on Cross-Atten | 5.93 | 0.37 | 0.520 | 0.80 | 61.70 | 0.25 | 1.31 | 0.27 |
| **Ours** | **4.50** | **0.35** | **0.55** | **0.83** | **75.74** | **0.28** | **1.49** | **0.30** |

## F ANALYSIS OF THE FINE-TUNED MODEL

To further verify that our method purely learns the customized attributes without compromising the original model and purely understands the target attributes, we solely use identical base text to evaluate whether our method can reconstruct the original portraits after fine-tuning. As shown in Fig. 13, standard fine-tuning markedly disrupts original response patterns, while our method maintains near-identical performance to the original model. For example, in the top-right case, the semantics of 'woman' is completely corrupted by standard fine-tuning, but we not only retain the character but also maintain high consistency in other attributes. The outstanding reconstruction

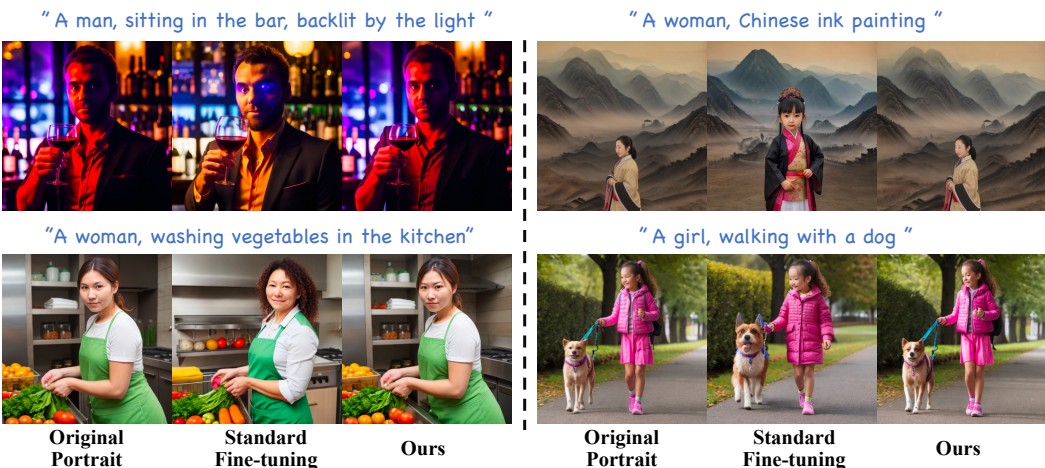

"A man, sitting in the bar, backlit by the light"     "A woman, Chinese ink painting"

"A woman, washing vegetables in the kitchen"     "A girl, walking with a dog"

| Original Portrait | Standard Fine-tuning | Ours | Original Portrait | Standard Fine-tuning | Ours |

Figure 13: **Reconstruction Results.** The three portraits for each case are generated by the fine-tuned model using only the same base text.

of portraits across varied scenes demonstrates our method's substantive retention of the original model's pre-trained prior knowledge.

## G    SENSITIVITY ANALYSIS OF LOSS

To determine the optimal settings for the three loss hyperparameters, we conducted a comprehensive sensitivity analysis. As shown in Fig. 14 The three segments of the plot correspond to the hyperparameters in Eq. 11 ($\lambda_1 \rightarrow \mathcal{L}_{M-text}$, $\lambda_2 \rightarrow \mathcal{L}_{M-fine}$, $\lambda_3 \rightarrow \mathcal{L}_{M-enhanced}$), demonstrating how FID scores vary with their values. Our analysis reveals that the optimal configuration occurs at $\lambda_1 = 0.2, \lambda_2 = 0.1, \lambda_3 = 0.6$, achieving the best FID score of 4.503 reported in our main results. It is noticed that the orange dashed line indicates the FID (4.013) of "Ours(w/o SFCM)" from Tab. 1, which exhibits over-alignment as visualized in Fig. 9.

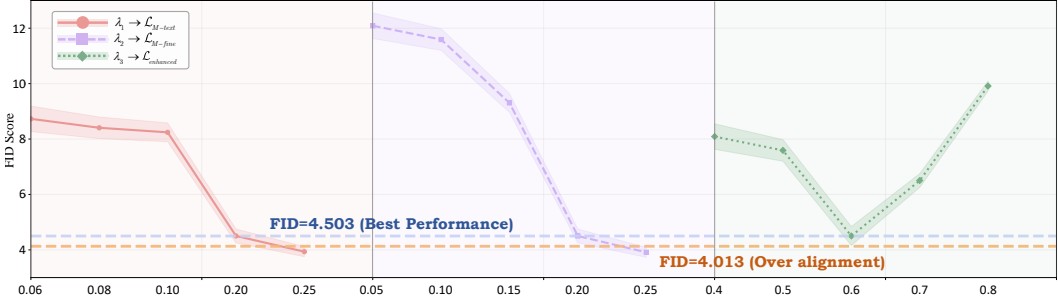

Figure 14: **Sensitivity analysis of three loss components** ($\lambda_1 \rightarrow \mathcal{L}_{M-text}$, $\lambda_2 \rightarrow \mathcal{L}_{M-fine}$, $\lambda_3 \rightarrow \mathcal{L}_{M-enhanced}$) **with respect to FID scores.**. FID varies with different parameter values for each loss component. FID=4.503 (optimal performance) and FID=4.013 (over-alignment with original model).

## H    ABLATION STUDY OF TRAINING STAGE

The main contribution of our method is the addition of an extra training stage on top of standard fine-tuning. To demonstrate the effectiveness of the two-stage training strategy, we conduct an ablation study on the training stages. As shown in Tab. 6, if only the second-stage contrastive learning is used, the model struggles to learn clean target attributes, resulting in significantly poor performance on

"CLIP-T." On the other hand, with only stage 1, the model is entirely affected by semantic pollution, failing to align with the original model behavior, thus performing worse on preservation metrics.

Table 6: **Ablation Study of the training Stage.**

| Method | Preservation | | | | | Responsiveness | Overall | |
|---|---|---|---|---|---|---|---|---|
| | FID (↓) | LPIPS (↓) | ID (↑) | CLIP-I (↑) | Seg-Cons (↑) | CLIP-T (↑) | HPSv2 (↑) | MPS(↑) |
| Only Stage-1 (Standard Fine-tuning) | 20.41 | 0.57 | 0.21 | 0.63 | 57.77 | 0.24 | 0.21 | 0.67 |
| Only Stage-2 | 7.18 | 0.38 | 0.13 | 0.71 | 63.72 | 0.19 | 0.24 | 1.12 |
| **Stage-1&2 (Ours)** | **4.50** | **0.35** | **0.55** | **0.83** | **75.74** | **0.30** | **0.28** | **1.49** |

# I   DISCUSSION ABOUT EDITING METHODS

As mentioned in the related work Sec. 2.3, incorporating text-driven editing methods (Deutch et al., 2024; Wang et al., 2024a; Kim et al., 2022; Ju et al., 2024) into the T2I model pipeline can produce similar results to ours. Here, we elaborate on the distinctions between our work and editing models and demonstrate that the improvement on inversion-based editing models when replacing their T2I model with ours.

The core distinction of our work lies in preventing additional textual concepts from disrupting T2I models, which fundamentally differs from I2I editing models that primarily focus on image manipulation through precise local modifications. Although the visual results of our method are presented in a pairwise comparison which may resemble those of editing work, the purpose is to demonstrate that our incremental learning approach preserves the integrity of the original model.

For an ideal AI-driven text-to-portrait creation, users aim for text to function like a brush in traditional painting, enabling targeted modifications to specific regions while preserving others unchanged. With existing technology, users can only achieve this by combining text-driven editing models, requiring: 1) Initial creation using a T2I model, 2) Refinement with an I2I editing model. However, in our framework, the T2I model can directly modify images via controlled text input during continuous generation, eliminating the need for additional I2I editing models. It can maintain consistency across continuous generations by preserving identical content for shared text elements. This makes the creative process more controllable, convenient, and aligned with intuition.

# J   DETAILS OF USER STUDY

We provide more details on our user study implementation. Besides qualitative and quantitative comparisons, we also conduct a user study to determine whether our method is preferred by humans. We invite 32 participants from different social backgrounds and each test session lasts about 30 minutes. During the investigation, as illustrated in Fig. 15, we conducted a pairwise comparison between our method and competitors across three key dimensions: 1) Original Behavior Consistency, 2) Target Attribute Response, and 3) Aesthetic Preference. For "Original Behavior Consistency", users were asked to select which of the two images better preserved consistency with the original model's outputs. For "Target Attribute Responsiveness", users evaluated which image more accurately reflected the target text description. For "Aesthetic Preference", users judged which image aligned better with their aesthetic preferences, considering factors such as visual quality and the absence of artifacts or distortions. This comprehensive evaluation framework ensures a thorough and objective assessment of our method's performance relative to existing approaches.

Figure 15: **The investigation page in user study.**

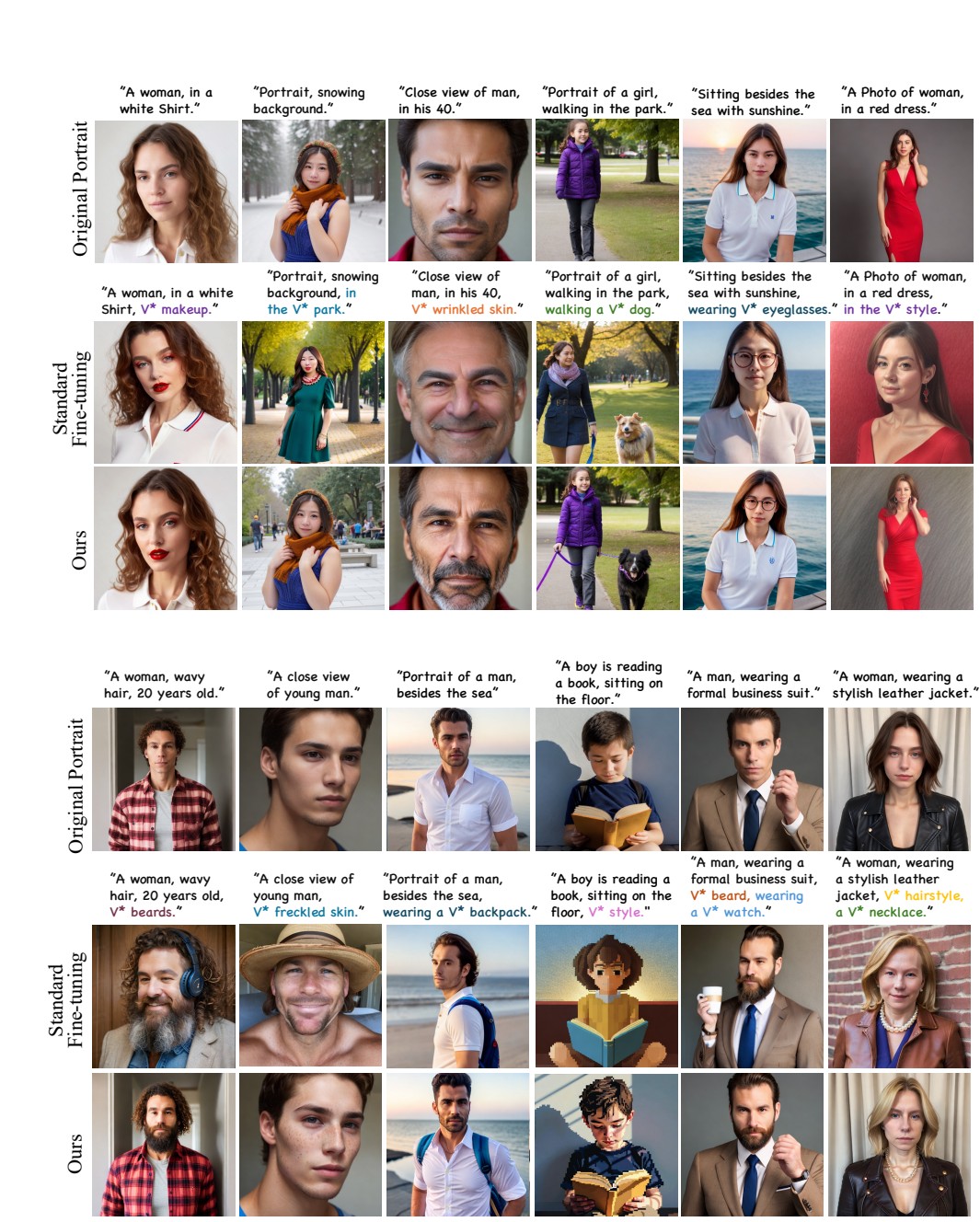

Figure 16: **More Results of SPF-Portrait in Pure Text-to-Portrait Customization.**

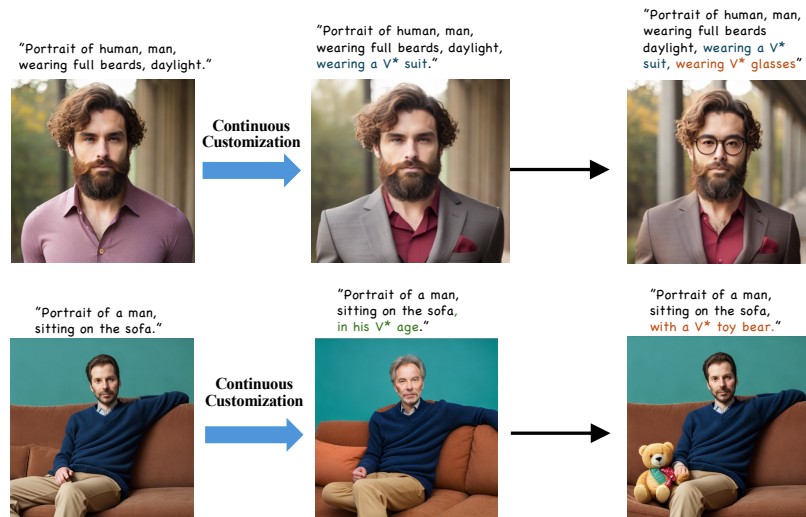

Figure 17: **More results of continuous replacements and additions of target semantics in text-to-portrait customization.** Our method demonstrates stable and excellent performance in continuous customization tasks, indicating its potential to play a role in the application scenarios of continuous AI portrait creation.

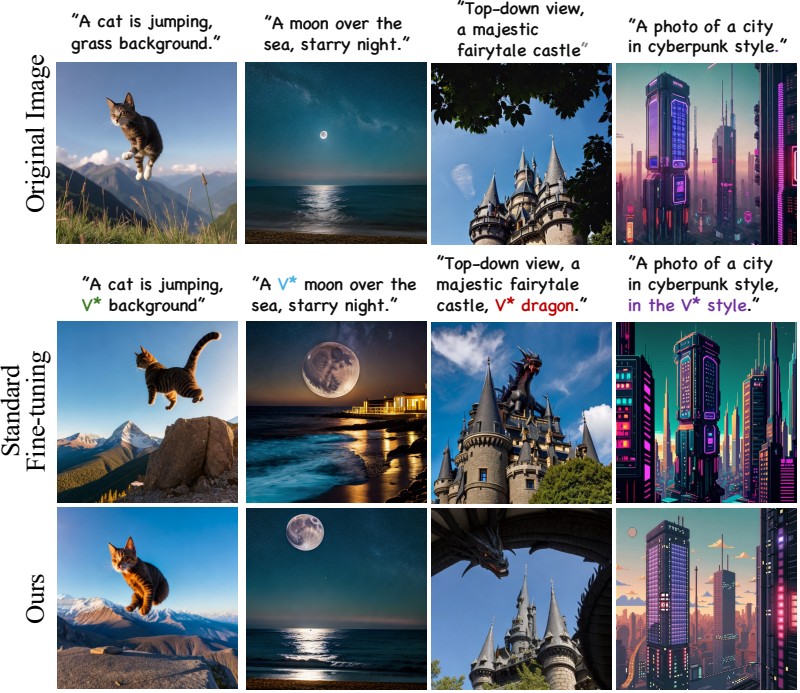

Figure 18: **More results of extending our method to the general Text-to-Image Customization.** These excellent experimental results demonstrate the feasibility of extending our method to the general T2I Customization. Our method has the potential to address the issue of semantic pollution in fine-tuning for the pure general T2I Customization.

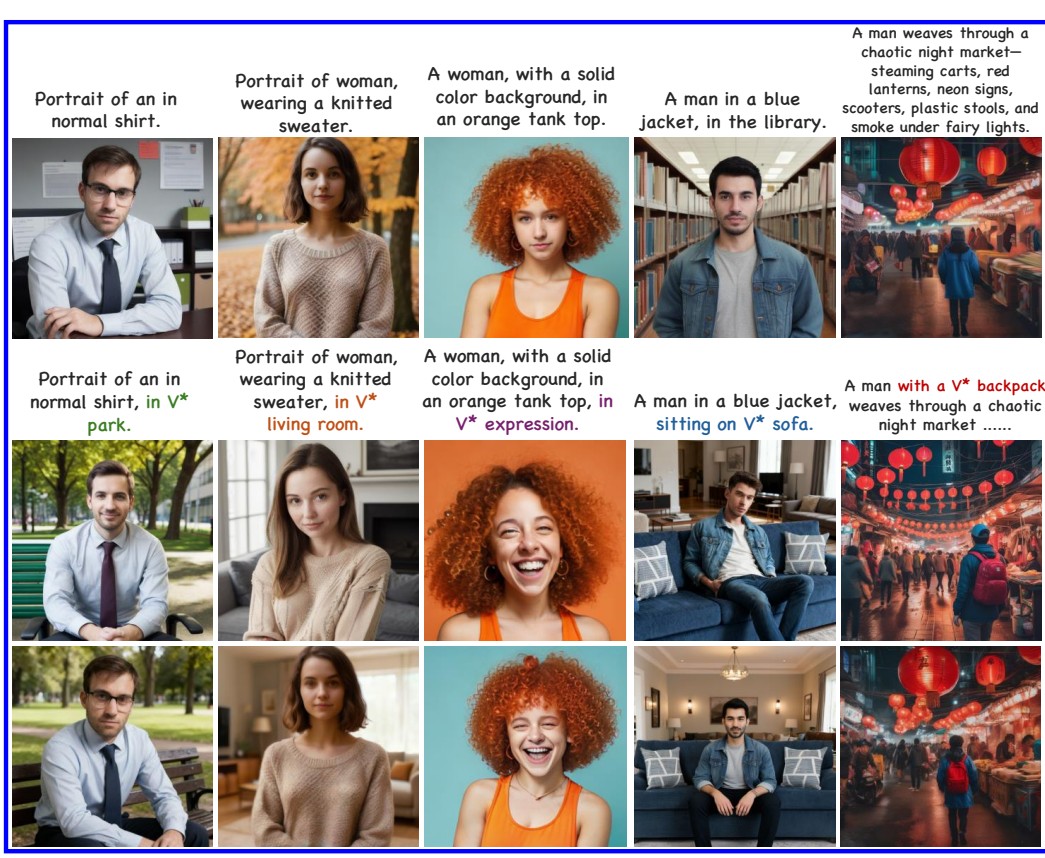

Figure 19: **More Results of SPF-Portrait in Pure Text-to-Portrait Customization.** We present results under various poses, expressions, backgrounds, and complex scenes.

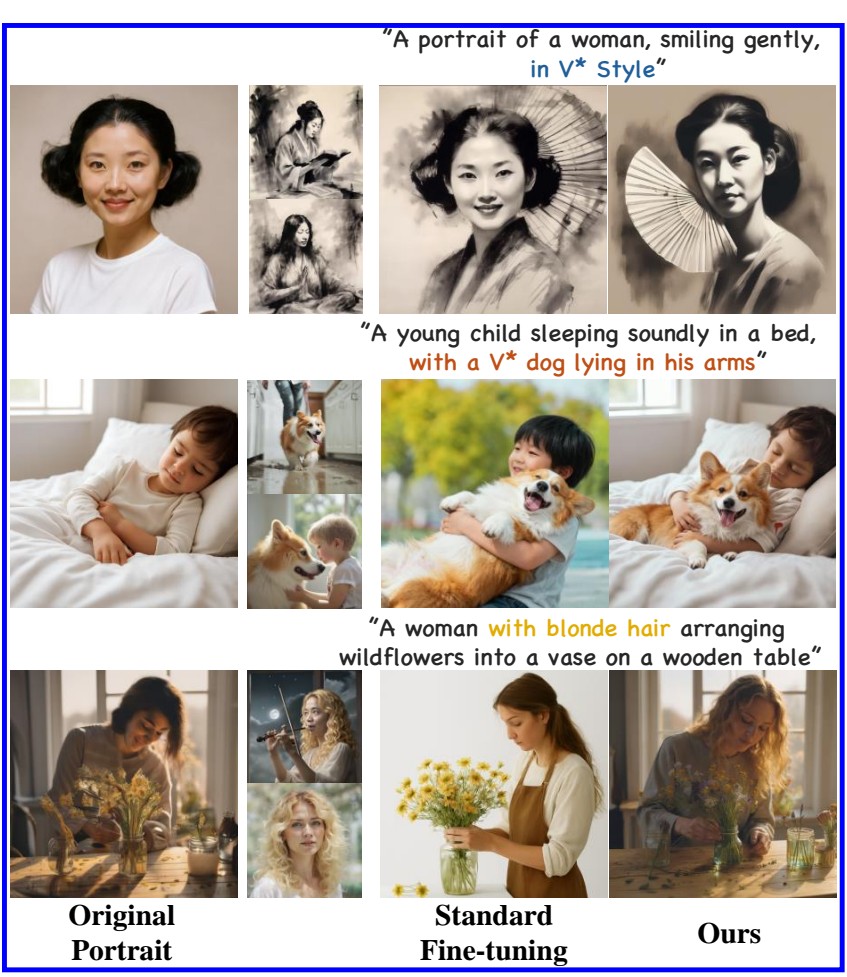

Figure 20: **Results of SPF-Portrait in DiT-based Diffusion Model (Stable Diffusion 3.5-Medium).**

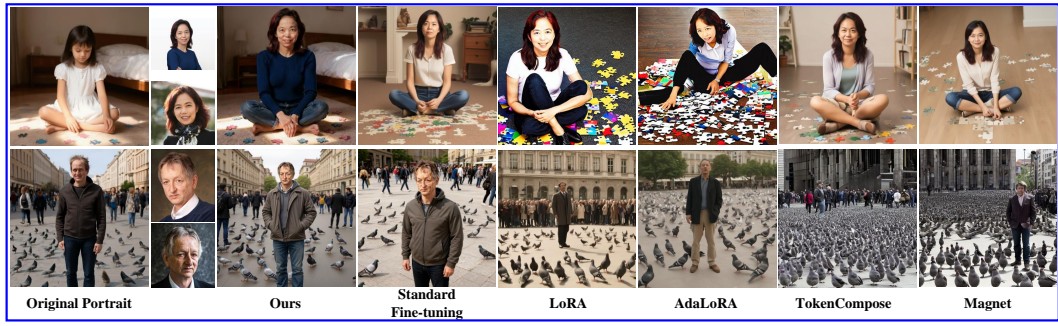

Figure 21: **Results of SPF-Portrait in ID Customization (Real Human Faces)**

