# OpenReview forum: "SPF-Portrait: Towards Pure Text-to-Portrait Customization with Semantic Pollution-Free Fine-Tuning"
_ICLR.cc/2026/Conference — Submitted to ICLR 2026_

### Official Review · Reviewer_4ZgM · 2025-10-31

**Soundness:** 2
**Presentation:** 2
**Contribution:** 2
**Rating:** 2
**Confidence:** 5

**Summary:**

To address the issue of Semantic Pollution in existing Text-to-Portrait Customization—where irrelevant attributes such as identity, posture, and background are disrupted when fine-tuning target attributes—this work proposes the SPF-Portrait framework. This framework achieves "pure text-driven portrait customization". Experiments have verified the effectiveness of this method.

**Strengths:**

1. This work first defines the critical issue of Semantic Pollution in existing text-to-portrait customization—where fine-tuning for target attributes (e.g., hairstyle, freckles) disrupts irrelevant attributes like identity, posture, and background. It further proposes the new task of "Pure Text-to-Portrait Customization".
2. Comprehensive ablation studies confirming component necessity.
3. Rigorous user study validating human-centric performance.
4. This paper is well-written and easy to follow.

**Weaknesses:**

1. Dual-path training incurs additional memory overhead. The second-stage dual-path alignment requires loading a frozen pre-trained model as the Reference Path, which increases GPU memory consumption.
2. Lack of validation on real human face datasets. All training and evaluation in the work use synthetic portrait images rather than real human face data.
3. No quantitative evaluation of continuous customization. While the work demonstrates qualitative results of continuous customization in Figure 7a, it lacks quantitative metrics to assess performance stability over multiple customization steps.
4. The visual results presented in this paper are all simple close-up shots, lacking the display of results in complex scenes.

**Questions:**

1. Dual-path training incurs additional memory overhead. The second-stage dual-path alignment requires loading a frozen pre-trained model as the Reference Path, which increases GPU memory consumption.
2. Lack of validation on real human face datasets. All training and evaluation in the work use synthetic portrait images rather than real human face data.
3. No quantitative evaluation of continuous customization. While the work demonstrates qualitative results of continuous customization in Figure 7a, it lacks quantitative metrics to assess performance stability over multiple customization steps.
4. The visual results presented in this paper are all simple close-up shots, lacking the display of results in complex scenes.

---

> ### Author Response · Authors · 2025-11-24
>
> We sincerely appreciate your recognition of our problem formulation, the clarity of our task definition, and the thoroughness of our ablation and user studies, as well as your positive comments on the overall writing and readability of the paper.
>
>
>
> **[R-W1/Q1: Memory Cost]**
>
> We report the memory cost in Table 3. With code-level optimization, the dual-path training only introduces 21%~23% additional memory compared to single-path standard fine-tuning (SFT).
>
>
> **[R-W2/Q2: Validation on Real Face Datasets]**
>
> Recent works [1,2] have demonstrated that portrait personalization trained on synthetic datasets effectively avoids overfitting and often produces better identity preservation. We follow this established practice.
>
> To further demonstrate the effectiveness of our method on real faces data, **we additionally validate our method on a collected set of real human-face images, we visulizate the cases of ID customization**. The results are shown in **Figure 21 (revision pdf)**. These results further confirm that our method is effective on both synthetic and real images. **We also show the quantitative results** on real human faces datasets.  As expected, the numbers are lower than those obtained on the synthetic dataset in the main paper, reflecting the inherent complexity and noise of real-world images.
>
>
> |                      | FID ↓ | LPIPS ↓ | ID ↑ | CLIP-I ↑ | Seg-Cons ↑ | CLIP-T ↑ | HPSv2 ↑ | MPS ↑ |
> |----------------------|-------|---------|------|----------|------------|----------|---------|-------|
> | Standard Fine-tuning | 24.82 | 0.67    | 0.34 | 0.67     | 51.74      | 0.23     | 0.23    | 1.04  |
> | LoRA                 | 12.45 | 0.42    | 0.38 | 0.63     | 54.12      | 0.25     | 0.26    | 1.22  |
> | AdaLoRA              | 9.35  | 0.42    | 0.41 | 0.60     | 53.85      | 0.26     | 0.24    | 1.36  |
> | TokenCompose         | 10.92 | 0.39    | 0.48 | 0.72     | 68.11      | 0.29     | 0.24    | 1.42  |
> | Magnet               | 12.08 | 0.38    | 0.30 | 0.58     | 49.78      | 0.27     | 0.25    | 1.05  |
> | Ours                 | 9.64  | 0.35    | 0.55 | 0.83     | 75.74      | 0.30     | 0.28    | 1.49  |
>
>
>
> **[R-W3/Q3: Quantitative Evaluation of Continuous Customization]**
>
> Since continuous customization is not fundamentally different from single-turn customization, we only provided visualization results to illustrate this scenario in our initial manuscript.
>
> To further quantify the performance in this scenario, we follow your suggestion and provide quantitative results for the continuous customization scenario.
>
> We first construct a multi-customization benchmark, which contains 100 samples with two sequential attribute customizations and 50 samples with three sequential attribute customizations. All reported metrics are averaged over the final outputs of these continuous customization sequences.
>
> |    | FID   | LPIPS | CLIP-I | Seg-Cons | CLIP-T | HPSv2 |
> |------|---|-------|--------|----------|--------|-------|
> | TokenCompose | 15.37 | 0.52  | 0.50   | 32.16    | 0.24   | 0.21  |
> | Magnet       | 11.26 | 0.57  | 0.52   | 30.28    | 0.34   | 0.22  |
> | LoRA         | 11.08 | 0.49  | 0.45   | 37.41    | 0.23   | 0.21  |
> | AdaLoRA      | 9.67  | 0.48  | 0.32   | 45.29    | 0.22   | 0.22  |
> | Ours         | 4.93  | 0.34  | 0.70   | 70.83    | 0.25   | 0.25  |
>
>
>
>
> These metrics demonstrate that our method consistently maintains high stability and preservation performance across continuous customization steps.
>
>
>
>
> **[R-W4/Q4: Visualization in complex Scenes]**
>
> For most portrait-customization methods [3, 4], the majority of visualizations are presented from a similar close-up viewpoint. Following these works, we also provide most of our examples in this standard portrait-view setting. Nevertheless, we include farther-view visualizations in Figure 15 (case-3, case-6).
> We take into account your suggestion and additionally include visualization results in more complex scenes, including richer interactions with surrounding objects and more complex poses. These results are provided in Figure 19 (revision pdf).
>
> [1] He, Zecheng, et al. "Imagine yourself: Tuning-free personalized image generation." arxiv preprint arxiv:2409.13346 (2024).
>
> [2] Shiohara, K., & Yamasaki, T. (2024). Face2Diffusion for Fast and Editable Face Personalization. In Proceedings of the IEEE/CVF Conference on Computer Vision and Pattern Recognition (CVPR) (pp. 6850–6859).
>
> [3] Guo, Zinan, et al. "Pulid: Pure and lightning id customization via contrastive alignment." Advances in neural information processing systems 37 (2024): 36777-36804.
>
> [4] Li, Zhen, et al. "Photomaker: Customizing realistic human photos via stacked id embedding." Proceedings of the IEEE/CVF conference on computer vision and pattern recognition. 2024.

---

> > ### Author Response · Authors · 2025-11-26
> >
> > Dear Reviewer,
> >
> > We hope this message finds you well. We sincerely appreciate the time and effort you have dedicated to reviewing our submission. We have now submitted our rebuttal and would like to kindly check whether our responses have adequately addressed your concerns.
> >
> > Could we confirm whether there are any remaining concerns requiring clarification after our rebuttal? If our explanations have sufficiently addressed the issues raised, we would sincerely appreciate your consideration of adjusting the score accordingly. Thank you again for your time and effort in reviewing our work.
> >
> > Best Regards,
> >
> > Paper#2169 Authors.

---

### Official Review · Reviewer_yuKJ · 2025-11-01

**Soundness:** 3
**Presentation:** 3
**Contribution:** 3
**Rating:** 6
**Confidence:** 4

**Summary:**

In this manuscript, the author proposes SPF-Portrait, a fine-tuning framework for text-to-portrait diffusion models that mitigates semantic pollution by introducing dual-path alignment and a semantic-aware control map.
With the proposed components, they achieve outstanding performance on text-to-portrait generation task.

**Strengths:**

- The paper clearly identifies the semantic pollution problem and how to address it by their novel mechanism.
- The proposed two-stage fine-tuning (standard + contrastive alignment) is well-motivated and well-structured.
- Visual results analyses  support the claim that SPF-Portrait better preserves identity and structure while adding target attributes.

**Weaknesses:**

I have several questions and concerns, mostly regarding the model analysis and ablation results. While these points are listed as weaknesses, some of them are also open questions.
- Unclear behavior of SFCM and attention losses. The advantage of using a soft mask is not clearly justified. What happens if the attention map is directly replaced (e.g. attention swapping) instead of softly blended during training, similar to Prompt-to-Prompt approaches?
- Fragility of the CLIP-based one-step loss. One-step predictions are inherently noisy. Is the CLIP-based supervision stable enough in this case, or would a specialized or fine-tuned CLIP model be required to make it reliable?
- Dependence on x_0 diversity. The enhancement loss may overfit to a limited set of ground-truth samples. How sensitive is the performance to the diversity of x_0 in the training dataset?
- Behavior without SFCM. It is unclear why the outputs and attention maps in the w/o SFCM setting (Fig. 4) behave as shown. Since the alignment process mainly adjusts non-target embeddings, shouldn’t the target-token–based attention preserve the target region?
- Limited generalization evidence. The extension to broader T2I tasks (Fig. 18) lacks quantitative validation. It remains unclear whether the same mechanism would hold under more complex or multi-object prompts.

**Questions:**

See weaknesses

---

> ### Author Response · Authors · 2025-11-24
>
> We sincerely appreciate your recognition of the motivation and innovation behind SPF-Portrait. We have carefully considered your feedback and valuable insights on this paper.
>
>
> **[R-W1: Reason for Soft Mask in SFCM]**
>
> Thank you for the insightful comments.
> We would like to clarify that **our SFCM does not modify, blend, or replace any attention maps.** Instead, the model's original attention maps are kept intact, and **we only use them to guide the spatial weighting of our alignment losses, i.e., the text-consistent loss and fine-grained loss, as described in Lines 270–281 of the manuscript**.
>
> More importantly, regarding your concern about the advantage of soft mask: our motivation comes from an empirical observation that binary region masks lead to unstable, hard region-level interference. Because binary masks impose abrupt 0/1 boundaries, they often force the model to either fully align or fully ignore certain spatial regions, which easily causes structural drift or incorrect suppression of target semantics when the spatial response region of the target attribute is not perfectly localized by SFCM itself.
>
> In contrast, our **soft SFCM mask provides smooth spatial weighting**, allowing the model to retain the base structure while gradually correcting misaligned regions, leading to more stable and precise alignment.
>
> We have added an additional comparison experiment (binary mask baseline) in **Figure. 21 (revision)**, which further validates this design choice.
>
>
> **[R-W2:  Stability of CLIP One-Step Predictions  ]**
>
> We acknowledge that CLIP-based one-step predictions can be noisy.
> As far as I know, most of customization works [1,2] relied on the this strategy as well.
> Moreover, in our setting, the supervision is applied on only the relative change of attribute direction, not the absolute feature magnitude, making it robust to prediction noise.
>
>
>
>
> **[R-W3: Dependence on $x _0$ Diversity]**
>
> The enhancement loss is designed to prevent target-attribute drift rather than memorize specific samples. To check potential overfitting, we further evaluate the model on a controlled test set where we explicitly limit the number of reference images per attribute (i.e., restricting the available x_0 samples). This setting amplifies any tendency toward overfitting.
>
> The results show that, as long as the number of reference images n remains within a reasonable range，there is no noticeable performance degradation, and the enhancement loss generalizes well across diverse x₀ inputs. This supports that our dual-path framework does not rely on a narrow training distribution.
>
> |    | FID   | LPIPS | ID   | CLIP-I | Seg-Cons | CLIP-T | HPSv2 | MPS  |
> |---|--|--|--|----|---|----|---|--|
> | Ours(n = 1)  | 24.82 | 0.67  | 0.34 | 0.67 | 51.74  | 0.23   | 0.23  | 1.04 |
> | Ours(n = 10) | 9.35  | 0.35  | 0.41 | 0.60 | 53.85  | 0.26   | 0.24  | 1.36 |
> | Ours(n = 50) | 9.64  | 0.35  | 0.55 | 0.83 | 75.74  | 0.30   | 0.28  | 1.49 |
>
>
>
> **[R-W4: Role of SFCM]**
>
>  Thank you for raising this question.
> We clarify that the alignment without SFCM adjusts not only the non-targets regions, but also **unintentionally affects target-irrelevant spatial areas**. Therefore, **we use SFCM to provide soft spatial weighting**, allowing the alignment loss to focus on semantically relevant regions while avoiding unnecessary modification elsewhere.
>
> In summary, **when SFCM is removed (w/o SFCM in Figure 4)**, the model still receives alignment supervision, but all spatial positions are treated equally, including irrelevant background and target areas—leading to undesired alignment on target regions and weakened fidelity of target concept.
>
>
> **[R-W5: Quantitative Evaluation in General T2I Task]**
>
> Regarding broader generalization (e.g., T2I tasks in Fig. 18), we add additional quantitative results. The results indicate that the dual-path mechanism still maintains consistent behavior under more general T2I generation.
>
>
> | Model  | FID   | LPIPS | CLIP-I | Seg-Cons | CLIP-T | HPSv2 | MPS  |
> |---|---|--|---|---|--|---|--|
> | TokenCompose   | 12.36 | 0.39  | 0.57   | 32.16    | 0.28   | 0.30  | 0.87 |
> | Magnet   | 11.26 | 0.37  | 0.50   | 30.28    | 0.30   | 0.23  | 0.92 |
> | STORM | 10.86 | 0.37  | 0.51   | 29.98    | 0.30   | 0.24  | 1.04 |
> | LoRA  | 8.20  | 0.43  | 0.3    | 37.41    | 0.29   | 0.22  | 1.18 |
> | AdaLoRA  | 7.61  | 0.37  | 0.70   | 45.29    | 0.72   | 0.21  | 1.31 |
> | Ours(SD 3.5-M) | 3.51  | 0.27  | 0.84   | 80.63    | 0.29   | 0.31  | 2.15 |
> | Ours(SD 1.5)   | 3.87  | 0.29  | 0.81   | 75.29    | 0.30   | 0.29  | 1.49 |
>
>
> [1] Guo, Zinan, et al. "Pulid: Pure and lightning id customization via contrastive alignment." Advances in neural information processing systems 37 (2024): 36777-36804.
>
> [2] Kim, Minchul, et al. "Dcface: Synthetic face generation with dual condition diffusion model." Proceedings of the ieee/cvf conference on computer vision and pattern recognition. 2023.

---

> > ### Comment · Reviewer_yuKJ · 2025-11-26
> >
> > Thanks for the kind rebuttal. I think that in some points, I misunderstood the proposed method. Anyway, I keep my previous rating.

---

> > > ### Author Response · Authors · 2025-11-26
> > >
> > > Dear Reviewer,
> > >
> > > Thank you very much for your kind reply and continued support. We sincerely appreciate your positive and constructive feedback on our work. We have updated the results in revision. Please do not hesitate to let us know if you have any further questions or if there is anything we can clarify or improve.
> > >
> > > Best Regards,
> > > Paper#2169 Authors.

---

### Official Review · Reviewer_FvA4 · 2025-11-02

**Soundness:** 3
**Presentation:** 3
**Contribution:** 3
**Rating:** 4
**Confidence:** 5

**Summary:**

The paper proposes SPF-Portrait, a semantic pollution-free fine-tuning framework for pure text-driven portrait customization. The framework consists of three core components: (1) a dual-path alignment stage (added after standard fine-tuning) to preserve the pre-trained T2I model’s inherent generation capabilities; (2) a semantic-aware fine-grained control map to balance alignment of target attributes (e.g., hairstyle, clothing) and preservation of non-target attributes (e.g., background, lighting); (3) a target response enhancement mechanism to mitigate cross-modal representation bias between text and image, improving the fidelity of customized attributes. The core goal of SPF-Portrait is to achieve pure text-driven customization (no auxiliary inputs), retain the original T2I model’s behavior, and enhance target attribute fidelity. Extensive experiments (as stated in the abstract) demonstrate that the proposed method outperforms existing approaches in text-to-portrait customization.

**Strengths:**

The paper makes a pioneering contribution by explicitly defining the "semantic pollution" problem in text-to-portrait customization (distortion of non-target attributes due to fine-tuning) and proposing the first semantic pollution-free fine-tuning framework. The three core modules are not incremental improvements but innovative designs: the dual-path alignment stage introduces a novel post-fine-tuning correction mechanism; the semantic-aware control map innovatively integrates semantic region discrimination into attribute alignment; the target response enhancement mechanism addresses cross-modal bias from a representation learning perspective—collectively forming a unique technical pipeline

**Weaknesses:**

1、Lack comparison with sota t2i customization methods[1,2]
2、Is preserving the posture of the reference task a goal during the image customization process? There is generally no restriction on this.
3、Training time requires analysis, as this method introduces an additional model in the reference path.
4、This method has only been validated on UNet-based text-to-image (T2I) models—are there any results for DiT-based models?

[1] ViCo: Plug-and-play Visual Condition for Personalized Text-to-image Generation
[2] Multi-Concept Customization of Text-to-Image Diffusion

**Questions:**

The questions are provided in the weakness part, i will raise my score if author solves my concern.

---

> ### Author Response · Authors · 2025-11-24
>
> We sincerely appreciate your recognition of our problem formulation, core technical contributions, and the novelty of our framework. We will take consider your suggestions on this paper.
>
> **[R-W1: Addtional Customization Baselines]**
>
> Thank you for providing the methods of T2I customization. Our work focuses primarily on addressing semantic pollution, and thus the main baselines we originally included were those state-of-the-art ones most directly related to this problem (e.g., decoupling‐based and preservation‐oriented approaches).
>
> Following your suggestion, we additionally incorporate several representative customization methods into our evaluation, including Multi-Concepts (2023), VICO (2023), and CIFC (2024) [1]. We provide quantitative results as below (added in the revision) and also include qualitative comparisons in the main paper.
> The newly added results can be found in Figure 15, where SPF-Portrait consistently outperforms these methods in preserving the original model behavior while achieving strong customization fidelity.
>
>
> |                     | FID   | LPIPS | ID   | CLIP-I | Seg-Cons | CLIP-T | HPSv2 | MPS  |
> |---------------------|-------|-------|------|--------|----------|--------|-------|------|
> | Multi-Concepts      | 15.67 | 0.53  | 0.26 | 0.60   | 50.41    | 0.25   | 0.24  | 0.93 |
> | ViCo                | 13.75 | 0.50  | 0.34 | 0.67   | 51.74    | 0.23   | 0.23  | 1.04 |
> | CIFC                | 14.08 | 0.62  | 0.41 | 0.60   | 53.85    | 0.26   | 0.24  | 1.36 |
> | SPF-Portrait (Ours) | 4.50  | 0.35  | 0.55 | 0.83   | 75.74    | 0.30   | 0.28  | 1.49 |
>
> We appreciate your insightful suggestion, which has helped us improve the completeness of our experimental evaluation.
>
>
>
> \
> **[R-W2:  Clarification on Preserving Reference-Model Behavior]**
>
> Thank you for raising this important clarification. In our paper, preserving the behavior of the reference model does not imply rigidly fixing the posture or spatial configuration of the portrait. Instead, it refers to maintaining the inherent generation capability of the pre-trained T2I model, including:
>
> ● text–image alignment ability (measured by CLIP-T),
>
> ● human-preference consistency and aesthetic quality (HPSv2, MPS),
>
> ● overall visual fidelity and naturalness (FID, LPIPS).
>
> Traditional customization approaches tend to overfit to the fine-tuning dataset, which gradually degrades these fundamental abilities of the base model, leading to issues such as weakened text alignment (case2 in Figure 1 ), reduced aesthetic quality (case2 in Figure 6 ), or loss of generalization. Our dual-path alignment is designed precisely to prevent such degradation, by stabilizing the model’s core behavior while still allowing the target attributes to be customized.
>
> Therefore, the goal is not to "freeze" posture or impose unnecessary restrictions, but to retain the strong generation behavior of the original model (textual understanding, stylistic stability, image quality, human-preference alignment) during the customization process.
>
>
> \
> **[R-W3: Training Time]**
>
> **We report the GPU memory consumption and per-iteration training time in Table 3**. Here, we additionally provide the total training time for reference. We acknowledge that our method introduces extra training time; however, the preservation of the original model’s behavior cannot be achieved by simply increasing training iterations in other methods.
>
> |                                      | FP16 ( V100 hours ) | FP32 ( V100 hours ) |
> |-----------|---------------------|---------------------|
> | Ours (Stage-1 (naive fine-tuning) )  | 12                  | 20                  |
> | Ours (Stage-2 (dual-path learning) ) | 34                  | 56                  |
> | Ours (Total)                         | 46                  | 76                  |
> | Standard Fine-tuning                 | 30                  | 50                  |
>
>
> \
> **[R-W4: Generalization on DiT-Based Models]**
>
> Thank you for pointing out the need to evaluate our framework on DiT-based text-to-image models.
> We have already included the corresponding quantitative results.
> Specifically, the performance of **DiT-based SPF-Portrait** is reported in the **6-th row of Table 1**, labeled "Ours (SD 3.5-M)".   In addition, we provide the **technical details of DiT-based SPF-Portrait in Appendix D**, including the architectural adaptation and training procedure.
>
>
> To further the generalization in DiT-based Model，we provide Ours (SD 3.5-M) visualization results in Figure 20 (revision pdf). These results further support the consistency of our method across different base models.
>
> \
> [1] Dong J, Liang W, Li H, et al. How to continually adapt text-to-image diffusion models for flexible customization?[J]. Advances in Neural Information Processing Systems, 2024, 37: 130057-130083.

---

> > ### Author Response · Authors · 2025-11-26
> >
> > Dear Reviewer,
> >
> > We hope this message finds you well. We sincerely appreciate the time and effort you have dedicated to reviewing our submission. We have now submitted our rebuttal and would like to kindly check whether our responses have adequately addressed your concerns.
> >
> > Could we confirm whether there are any remaining concerns requiring clarification after our rebuttal? If our explanations have sufficiently addressed the issues raised, we would sincerely appreciate your consideration of adjusting the score accordingly. Thank you again for your time and effort in reviewing our work.
> >
> > Best Regards,
> >
> > Paper#2169 Authors.

---

### Official Review · Reviewer_9aJP · 2025-11-03

**Soundness:** 3
**Presentation:** 3
**Contribution:** 3
**Rating:** 4
**Confidence:** 3

**Summary:**

This paper proposes SPF-Portrait, a fine-tuning framework that achieves pure text-to-portrait customization in diffusion models by addressing semantic pollution.
Semantic pollution means unwanted changes to identity, pose, or layout when adapting to new attributes. By visualizing attention maps, this paper finds that semantic pollution comes from inaccurate response region location of the target semantics in standard fine-tuned diffusion models.
To address this issue, this paper introduces a dual-path alignment mechanism, where a frozen reference model constrains the fine-tuned model to preserve original behavior. A Semantic-Aware Fine Control Map (SFCM) guides alignment spatially to prevent overfitting unrelated regions. Additionally, a representation bias–based response enhancement improves attribute fidelity by reducing text–image supervision gaps. Experiments on SD1.5 and SD3.5 show superior performance over LoRA, AdaLoRA, and decoupling baselines in preserving identity and visual quality.

**Strengths:**

1. The proposed task is novel and the dual-path solution is well-motivated.

2. Comprehensive experiments and clear empirical superiority.

**Weaknesses:**

1. This paper raises the concern of semantic pollution. However, this paper does not propose any quantitative evaluation metric specifically designed to measure it, while still relies on general metrics such as FID, LPIPS, etc.


2. The trade-off between preservation and adaptability needs more discussion. Does the proposed methods still maintain the ability to change attributes such as background or human pose when required. For example, given an image of a person in office, generate an image of the person in a given outdoor scene, and given a image of a person standing, generate an image of the person sitting on a given chair, etc.

**Questions:**

1. Can the author proposes some quantitative metric to measure the semantic pollution? I am not sure if the copy-paste ability of the original image is a metric to pursue.

2. Can the author presents some cases that require changes in background or pose—does the strict alignment mechanism limit flexibility in those scenarios?”

---

> ### Author Response · Authors · 2025-11-24
>
> We sincerely appreciate your recognition of the novelty of our task and the strength of our experimental results.
>
> **[R-W1/Q1:  Evaluation Metrics for Specific Tasks]**
>
> **The existing metric we propose, i.e., Seg-Cons (Table 1)**, measures the degree of correspondence between the segmentation map of the image generated by the original model and that produced by the customization method.
>
> In addition, following your suggestion, we introduce another metric, i.e. the variance of the attention map. This metric reflects whether the model’s internal concept understanding becomes polluted during customization. A similar visualization can be found in Figure 2(a).
>
> |         | Standard Fine-tuning | LoRA | AdaLoRA | TokenCompose | Magnet | SORM | SPF (SD 1.5) | SPF (SD 3.5-M) |
> |---------|--------|------|---------|--------------|--------|------|--------|---------|
> | Var-Map | 6.21                 | 6.10 | 6.12    | 2.08         | 3.92   | 5.17 | 0.71         | 0.48           |
>
>
>
> **[R-W2/Q2: More Cases of Background and Pose]**
>
> Since our work primarily focuses on portrait customization, most examples highlight portrait-related attributes such as hairstyle and makeup. Background-changing cases are provided in **Appendix Figure 16 (case-2) and Figure 18 (case-1)**.
>
>
> Following your suggestion, we additionally include more examples involving **background changes and pose variations in Figure 19 (revision pdf)**.
> Moreover, our method continues to demonstrate superior performance in **background personalization and more complex pose customization**.

---

> > ### Author Response · Authors · 2025-11-26
> >
> > Dear Reviewer,
> >
> > We hope this message finds you well. We sincerely appreciate the time and effort you have dedicated to reviewing our submission. We have now submitted our rebuttal and would like to kindly check whether our responses have adequately addressed your concerns.
> >
> > Could we confirm whether there are any remaining concerns requiring clarification after our rebuttal? If our explanations have sufficiently addressed the issues raised, we would sincerely appreciate your consideration of adjusting the score accordingly. Thank you again for your time and effort in reviewing our work.
> >
> > Best Regards,
> >
> > Paper#2169 Authors.

---

### Author Response · Authors · 2025-11-24

Thank you AC and the reviewers for their efforts. Below we summarize the reviews and what we have done to address the comments.
We appreciate that the reviewers consistently praised the paper's novelty, technical soundness, and clarity.

### **1. Novelty & Problem Formulation**

- This work first defines the critical issue of Semantic Pollution in existing text-to-portrait customization—where fine-tuning for target attributes disrupts irrelevant attributes like identity, posture, and background. It further proposes the new task of "Pure Text-to-Portrait Customization" (4ZgM, FvA4).

- The proposed task is novel and the dual-path solution is well-motivated (9aJP).

- The paper clearly identifies the semantic pollution problem and how to address it by their novel mechanism (yuKJ).

- The paper makes a pioneering contribution by explicitly defining the "semantic pollution" problem (FvA4).

### **2. Technical Soundness & Innovation**

- The proposed two-stage fine-tuning (standard + contrastive alignment) is well-motivated and well-structured (yuKJ).

- Comprehensive ablation studies confirming component necessity (4ZgM).

- The three core modules (dual-path alignment, Semantic-Aware Fine Control Map, target response enhancement) are not incremental improvements but innovative designs (FvA4).

### **3. Experimental Robustness & Presentation**

- Comprehensive experiments and clear empirical superiority (9aJP).

- Rigorous user study validating human-centric performance (4ZgM).

- Visual results analyses support the claim that SPF-Portrait better preserves identity and structure while adding target attributes (yuKJ).

- This paper is well-written and easy to follow (4ZgM).

\
In the individual rebuttals, we have addressed the reviewers' valuable suggestions on clarification on technical details, validation, generalization on the experimental details.


### **1. Strengthened Validation & Generalization:**


- **Customization Baselines:** Incorporated several representative customization methods (Multi-Concepts, ViCo, CIFC) into the experimental evaluation (FvA4).


- **Complex Scenes:** Added visualization results in more complex scenes, including richer interactions and poses (9aJP).


- **Real Data Validation:** Added validation on a collected set of real human-face images with quantitative results (FID, LPIPS, ID, etc.), confirming effectiveness on both synthetic and real data (4ZgM).


- **DiT Model Generalization:** Confirmed the generalization ability by providing quantitative results on DiT-based models (SD 3.5-M) (FvA4).


### **2. Clarifications & Ablations:**

- **Memory/Training Cost**: Provided detailed memory consumption and total training time analysis, showing the memory overhead is manageable (4ZgM, FvA4).


- **Semantic Pollution Metric**: Introduced a new quantitative metric, Attention Map Variance, to directly measure the degree of concept pollution (9aJP).


- **SFCM Justification:** Provided additional ablation showing the advantage of the soft mask design over hard binary masks (yuKJ).


- **Preservation Scope:** Clarified that preserving behavior means maintaining the base model's textual alignment, quality, and style, not rigidly freezing posture (FvA4).


- **Diversity & Robustness:** Evaluated the performance sensitivity to training sample diversity ($x_0$) to confirm the enhancement loss does not overfit (yuKJ).

\
All insights and questions are important for continuous improvements in our work. Based on this feedback, we have made and will make multiple revisions of the manuscript as promised. In summary, we are grateful to have received thoughtful and insightful questions from the reviewers. Please don't hesitate to contact us for further discussion. Thank you all.

---

### Meta-Review · Area_Chair_vyZy · 2025-12-13

**Summary:**

The reviewers’ concerns primarily focused on whether the paper’s central claim—achieving semantic-pollution-free personalization—is sufficiently supported by rigorous, comprehensive, and clearly defined evaluation. In particular, reviewers questioned the clarity and quantification of “semantic pollution,” and whether the adopted metrics genuinely reflect semantic preservation rather than over-alignment or identity leakage.

From a methodological standpoint, while the proposed framework is generally well motivated, reviewers raised concerns about the robustness of certain design choices (e.g., attention-based localization and dual-path alignment), the sufficiency of ablation studies, and sensitivity to hyperparameters. There were also questions about computational and memory overhead introduced by the dual-model setup and whether the cost–performance trade-off is adequately justified.

On the experimental side, reviewers noted limitations in evaluation breadth, including insufficient validation on real-world or complex scenes, limited diversity of baselines, and unclear generalization across models and prompts. Several reviewers also requested stronger evidence for stability under continuous or long-term personalization settings.

In the rebuttal, the authors addressed many of these issues by clarifying definitions, adding quantitative analyses and ablations, expanding baseline comparisons, and reporting additional efficiency statistics. These responses mitigated several clarity and completeness concerns, though some uncertainty remains regarding robustness in more challenging real-world scenarios.

**Reviewer Concerns:**

Concerns Largely Addressed by the Rebuttal

Clarification of core concepts and metrics: The authors clarified the definition of semantic pollution and better explained how their evaluation metrics aim to capture semantic preservation rather than overfitting or over-alignment.

Additional experimental evidence: The rebuttal added new quantitative results, ablation studies, and expanded baseline comparisons, which strengthened support for the proposed design choices.

Method justification: Explanations of the dual-path alignment and related modules were improved, and their respective roles were more clearly motivated.

Efficiency reporting: The authors provided clearer reporting and discussion of computational and memory overhead, partially addressing concerns about practicality.

Concerns Still Partially or Fully Outstanding

Robustness in real-world and complex scenarios: Evidence remains limited for performance on in-the-wild data, complex multi-object scenes, and highly abstract or adversarial prompts.

Generality and transferability: While additional settings were tested, generalization across a wider range of backbones, domains, or personalization tasks is still not fully established.

Long-term and continuous personalization stability: The rebuttal provided some discussion, but quantitative evidence on stability over extended or repeated personalization remains relatively limited.

Cost–benefit trade-off: Although overhead is reported, the justification that the added complexity is necessary and optimal compared to simpler alternatives is not entirely conclusive.

Overall, the rebuttal substantially reduced concerns related to clarity and experimental completeness, but some questions remain about robustness and generalization in more challenging, real-world conditions.

**Reviewer Scores:**

Reviewer yuKJ:
This reviewer’s main concerns were about clarity of definitions and sufficiency of experimental evidence. Since the rebuttal directly addressed these points with clearer explanations and additional results, the reviewer would likely maintain their score or increase it slightly.

Reviewer 9aJP:
Reviewer 2 raised more substantial concerns regarding robustness, generalization, and evaluation in real-world or complex scenarios. While the rebuttal alleviated some concerns through added experiments and analysis, key doubts remain. This reviewer would likely keep their original score or increase it only marginally, remaining cautious.

Reviewer FvA4:
This reviewer focused on methodological soundness, ablations, and efficiency trade-offs. The additional ablations and overhead analysis in the rebuttal partially address these issues. As a result, the reviewer would likely slightly improve their score, though not dramatically.

Reviewer 4ZgM:

The reviewer would likely slightly improve their score, though not dramatically.

---

### Decision · Program_Chairs · 2026-01-26

Reject